# Structural insight into Tn3 family transposition mechanism

Alexander V. Shkumatov [1,2,4], Nicolas Aryanpour [3], Cédric A. Oger [3], Gérôme Goossens[3,5], Bernard F. Hallet [3] ✉ & Rouslan G. Efremov[1,2] ✉

Transposons are diverse mobile genetic elements that play the critical role as genome architects in all domains of life. Tn3 is a widespread family and among the first identified bacterial transposons famed for their contribution to the dissemination of antibiotic resistance. Transposition within this family is mediated by a large TnpA transposase, which facilitates both transposition and target immunity. Howtever, a structural framework required for understanding the mechanism of TnpA transposition is lacking. Here, we describe the cryo-EM structures of TnpA from Tn4430 in the apo form and paired with transposon ends before and after DNA cleavage and strand transfer. We show that TnpA has an unusual architecture and exhibits a family specific regulatory mechanism involving metamorphic refolding of the RNase H-like catalytic domain. The TnpA structure, constrained by a double dimerization interface, creates a peculiar topology that suggests a specific role for the target DNA in transpososome assembly and activation.

Through their ability to mobilize and rearrange DNA sequences, transposons correspond to an inexhaustible source of genetic alterations[1,2], such as de novo creation of genes, the establishment of regulatory networks, exchange of genetic material by horizontal transfer[3,4], and the emergence and spread of antibiotic resistances[5–8]. Among these, Tn3-family transposons were the earliest bacterial transposons identified, owing to their implication in the transmission of ampicillin resistance[9]. Since then, numerous studies have isolated members of the Tn3-family from virtually all bacterial groups, where they act as mobile platforms for a variety of passenger genes, including those conferring resistance to multiple classes of antibiotics[8,10] (Supplementary Fig. 1). Notably, Tn3 transposons family were shown to be involved in the recent outbreak of carbapenem-resistant enterobacteria and in the dispersal of colistin resistance, wherein the use of these two antibiotics is often recognized as the "last-resort"[11–15].

Central to the efficiency of these transposons is their replicative "paste-and-copy" transposition mechanism in which duplication of the transposon occurs along with its integration into the target DNA[10,16]. Transposition is initiated by the transposase TnpA, an unusually large

member (~1000 amino acids) of the DDE/D superfamily of nucleotidyl transferases. TnpA cleaves the 3′-ends of the transposon and joins them to the target using a conserved RNase H-like domain[10,17–19] (Supplementary Fig. 2). This generates a strand transfer product that is then processed by the host replication system, producing two copies of the transposon (Fig. 1a). The reaction proceeds through the formation of a nucleoprotein complex, transpososome, which brings together the whole donor molecule carrying the transposon and the target (Fig. 1a). This distinguishes the "paste-and-copy" mode of transposition from the "cut-and-paste" and "copy-out-paste-in" mechanisms used by other DDE/D transposases, in which the element detaches completely from the donor prior to its integration into the target[16,20,21]. TnpA is the only known transposase that facilitates both transposition and target immunity[10], wherein target immunity is an intriguing regulation mechanism that prevents multiple transposon insertions into the same target and is believed to prevent self-destruction[22–28].

Despite their relevance in biological systems, the transposition mechanism of Tn3-transposons family is poorly understood. Recent

[1]Center for Structural Biology, Vlaams Instituut voor Biotechnologie, Brussels, Belgium. [2]Structural Biology Brussels, Department of Bioengineering Sciences, Vrije Universiteit Brussel, Brussels, Belgium. [3]Louvain Institue of Biomolecular Science and Technology, Université Catholique de Louvain (UCLouvain), Croix du Sud 4/5, 1348 Louvain-la-Neuve, Belgium. [4]Present address: Confo Therapeutics, Brussels, Belgium. [5]Present address: Thermo Fisher Scientific, Seneffe, Belgium. ✉e-mail: bernard.hallet@uclouvain.be; rouslan.efremov@vub.be

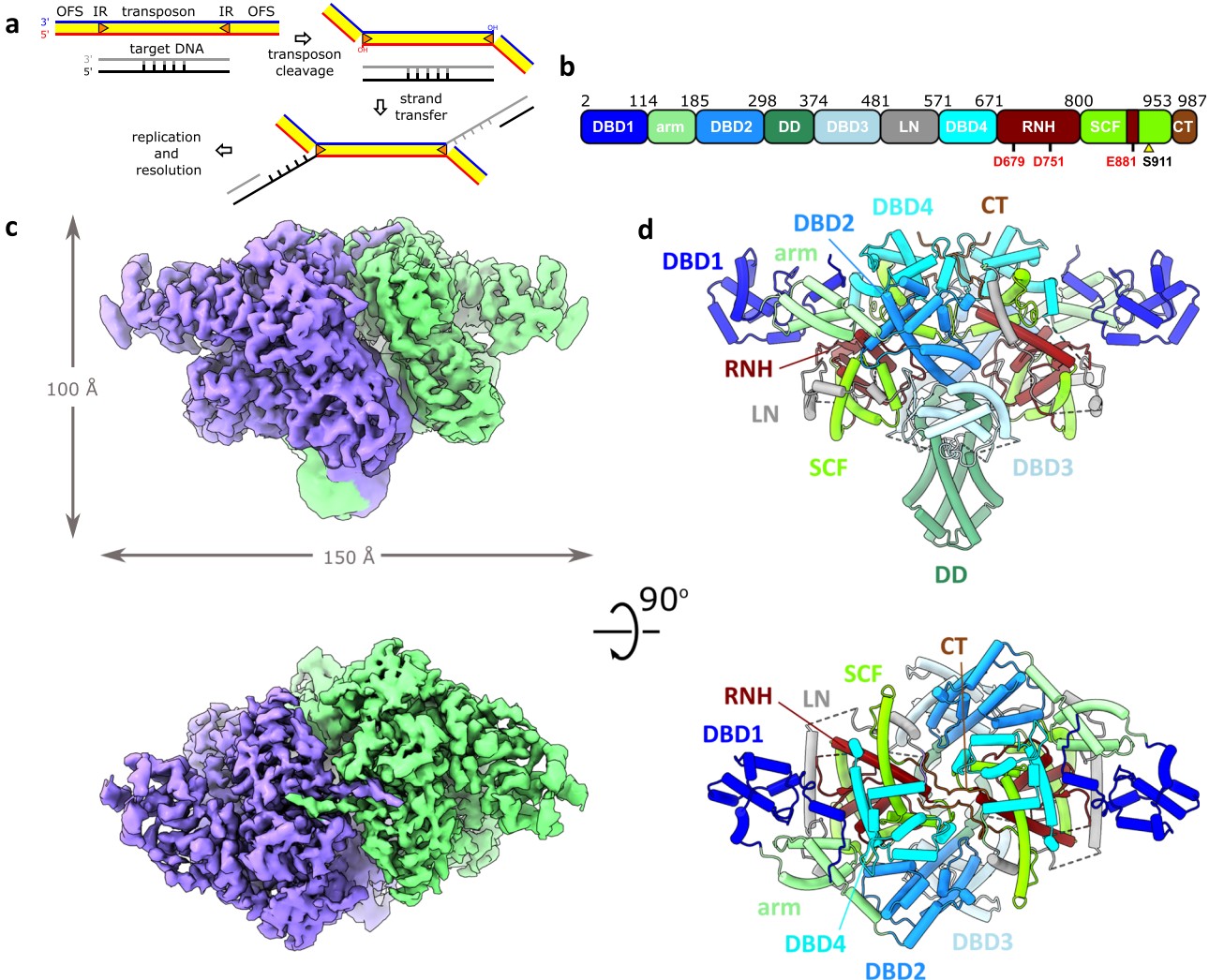

Fig. 1 | Function and architecture of TnpA. a Schematic of paste-and-copy transposition pathway of TnpA. b Linear diagram showing subdivision of TnpA into structural domains. DBD1-4 DNA binding domains 1–4, DD dimerization domain, LN linker domain, RNH RNase H-like catalytic domain, SCF scaffold domain, CT C-terminal tail. Positions of the catalytic DDE triad and activating mutation are indicated. c Cryo-EM map of TnpA$^{WT}$ in apo conformation. The map is colored by protomer. d Structure of TnpA$^{WT}$ in apo conformation shown in cartoon representation and colored by structural domains. The domains are color coded as in panel (b). Domains corresponding to one protomer are labeled.

studies on the Tn3-family member, Tn4430, have laid the foundation in unraveling this mechanism through the characterization of gain-of-function TnpA mutants defective in target immunity[28,29]. However, in the absence of structural information, the molecular interpretation of the data remained very sketchy.

In this study, we determined the single-particle cryo-EM structures of Tn4430 TnpA from *Escherichia coli*, and its cryo-EM structure was solved to an average resolution of 3.6 Å (Fig. 1, Supplementary Table 1, Supplementary Figs. 3, 4). Using density of the apo form and complexes of TnpA with IR ends (see below), an ab initio model containing 92% of the 987 residue-long protein was built (Supplementary Table 2).

The cryo-EM structure of apo TnpA$^{WT}$ revealed that it exists as a dimer, in contrast with the previously proposed model[29] (Fig. 1c, d). Each TnpA protomer can be divided into ten predominantly α-helical

## Results

### Architecture of apo TnpA
The wild-type TnpA (TnpA$^{WT}$) was expressed in *Escherichia coli*, and its cryo-EM structure was solved to an average resolution of 3.6 Å (Fig. 1, Supplementary Table 1, Supplementary Figs. 3, 4). Using density of the apo form and complexes of TnpA with IR ends (see below), an ab initio model containing 92% of the 987 residue-long protein was built (Supplementary Table 2).

The cryo-EM structure of apo TnpA$^{WT}$ revealed that it exists as a dimer, in contrast with the previously proposed model[29] (Fig. 1c, d). Each TnpA protomer can be divided into ten predominantly α-helical

structural domains (Fig. 1b, d). They are arranged in a 140 Å long stem and a disc-shaped protrusion with a diameter of 50 Å in the middle of the stem. The stem is composed of four DNA-binding domains (DBD) (see below), an α-helical arm domain that separates DBD1 from DBD2, DBD3, and DBD4 by ~40 Å, and dimerization domain (DD) which is poorly ordered in apo state but well-resolved in complexes with DNA (see below) (Figs. 1d, 2a, b). The protrusion is composed of a 90 amino acid long linker (LN) bridging DBD3 and DBD4, and a catalytic RNase H-like (RNH) domain. The RNH domain is encircled by α-helices referred to as the scaffold domain (SCF, Fig. 1c, Supplementary Movie 1). SCF is composed of a TnpA-specific RNase H insertion domain[21,30–34] (Supplementary Fig. 5) and α-helical structures downstream of RNH. Overall, TnpA bears an architecture different from other structurally characterized transposases, wherein besides DBD1, DBD4, and RNH domains, it also contains small domains with novel folds (Supplementary Table 3).

### Architecture of TnpA-transposon end complexes
Our attempts to obtain a paired-end complex (PEC)[29] between TnpA$^{WT}$ and two Tn4430 terminal inverted repeats (IR) resulted in the generation of only a minor fraction of TnpA-DNA complexes

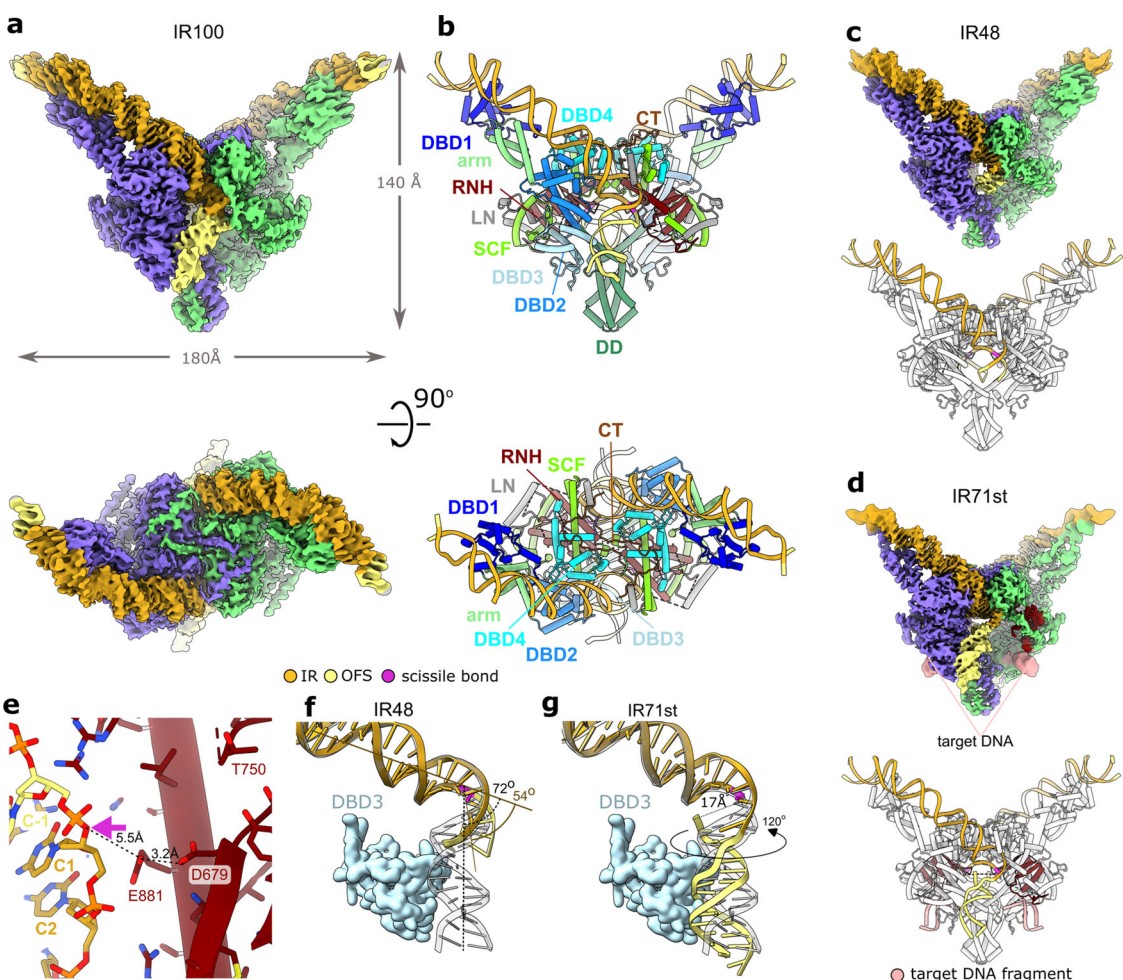

**Fig. 2 | Structure of TnpA^S911R with DNA substrates. a** Cryo-EM map of TnpA^S911R in complex with IR100 substrate. The map is colored by protomers. The DNA is colored by functional regions as indicated in the visual legend. **b** Structure of TnpA^S911R-IR100 complex shown in cartoon representation and colored by structural domains. The domains are color coded as in Fig. 1. Domains corresponding to one protomer are labeled. **c**, **d** Cryo-EM maps and structure in cartoon representation for TnpA^S911R-IR48 and TnpA^S911R-IR71st complexes, respectively. For clarity, the protein is shown in light-gray, and DNA is colored as in panel (**b**). In panel (**d**) the RNH domain is colored in bordeaux to show its interaction with DNA branch mimicking target DNA (light rose). **e** Details of RNH-trans DNA interactions. DNA is color coded as in panel (**b**). The scissile bond is indicated with a pink arrow. Resolved catalytic residues of the RNH domain are labeled. Indicative distances between the scissile bond and the catalytic residues are shown. D751 is not resolved but preceding it residue T750 was built in the low-resolution density. **f**, **g** Conformational differences between outer flanking segment (OFS) for IR48 (**f**) and IR71st (**g**) relative to IR100 substrate shown as transparent gray. The structure of DBD3 relative to which the substrates move is shown as space filling model.

(Supplementary Fig. 4b, c). Therefore, we used a previously identified hyperactive and immunity-deficient mutant S911R (TnpA^S911R)[28,29]; this enabled us to solve the structures of TnpA in complex with two linear IR substrates IR100 and IR48, which contained 38 base pair (bp) long TnpA recognition sequences of the IR end[10] flanked on each side with 31 and 5 bps corresponding to the inner and outer flanking segments, respectively (Supplementary Fig. 3a). These structures corresponded to the PEC and are referred to as TnpA^S911R-IR100 and TnpA^S911R-IR48, respectively. We also solved the structure of the strand transfer-like complex of TnpA^S911R with the substrate IR71st, branched at the IR 3′-end cytosine (Supplementary Fig. 3a), TnpA^S911R-IR71st. It mimics the strand transfer product of a 3′-cleaved IR end into the target DNA. The structures were solved to a resolution between 2.9 and 3.1 Å (Fig. 2, Supplementary Figs. 6–8, Supplementary Table 1, Supplementary Movie 1) and allowed modeling of up to 54 bp long DNA substrate comprising the complete IR sequence and fragments of inner and outer flanking DNA.

The binding of transposon ends is accompanied by large conformational changes that transform the compact apo form into an expanded V-shaped structure with ~140 Å long edges (Fig. 2a–c,

Supplementary Movie 1). In all IR-bound complexes, the structures of dimeric TnpA^S911R were virtually identical (Fig. 2a–d). The bound IR sequence curves smoothly, while the outer flanking segment bends sharply at the TnpA cleavage site (Fig. 2a, b). The scissile phosphate is positioned at the center of the dimer, where it is exposed to the RNase H-like domain of another TnpA protomer (Fig. 2b, e). This cis-trans arrangement, wherein one subunit recognizes and binds to one transposon end in cis (cis-interaction and cis-DNA), while catalysing DNA cleavage and strand transfer in trans on the partner end (trans-interaction and trans-DNA), is a convergent feature of most characterized DDE/D transposases despite their structural heterogeneity[21,30–32,35].

Protein–protein dimerization interfaces were very similar between the apo and DNA-bound conformations and were associated with two distinct areas within each protomer: DD and a C-terminal tail. DD protrudes out of the dimer (Figs. 1, 2, Supplementary Movie 1) as an extended α-helical bundle, in which three tightly packed α-helices contributed by each protomer primarily form hydrophobic interactions with contact area of ~1600 Å² (Supplementary Fig. 9a, Supplementary Table 4). In the apo form, DD is flexible, but low-resolution

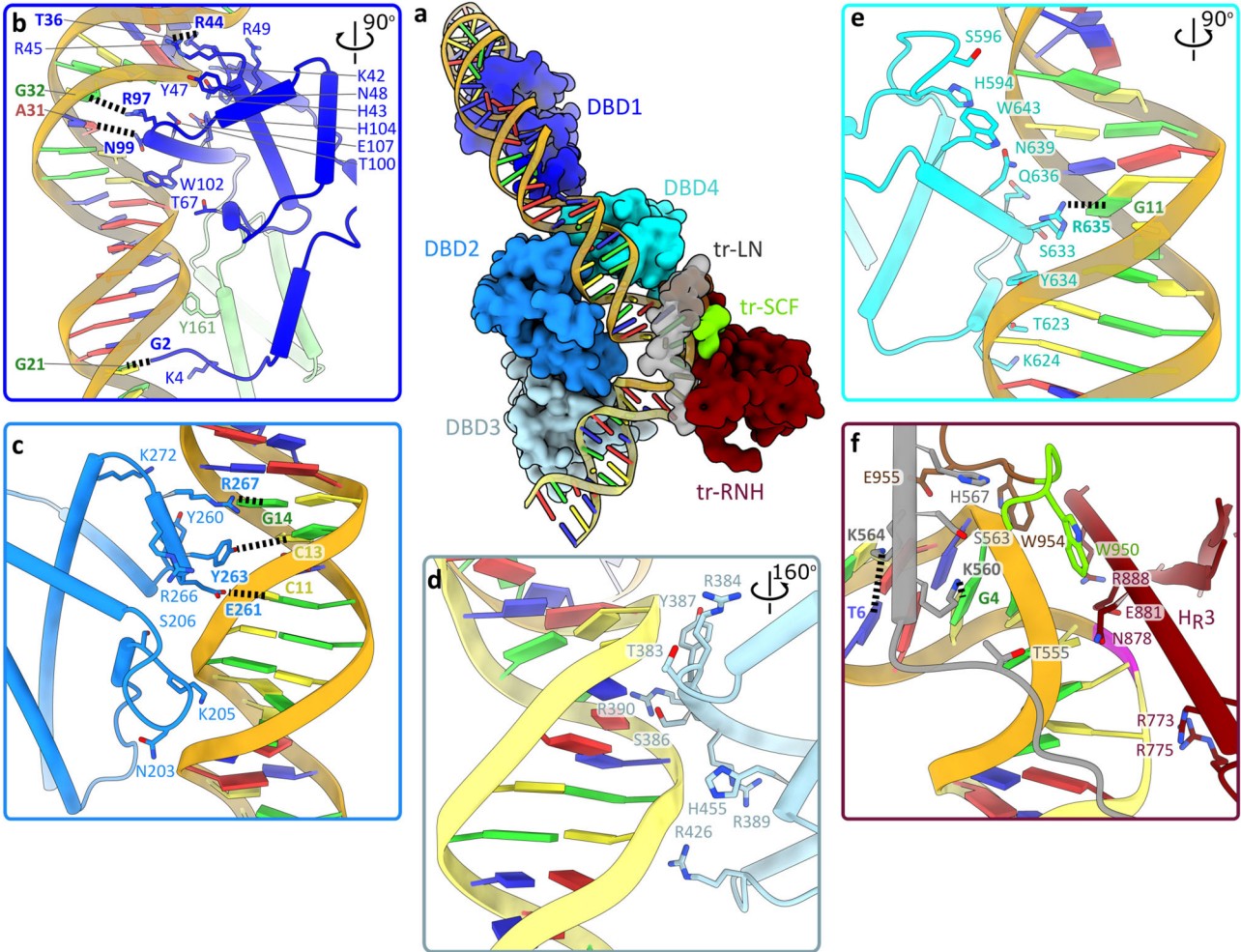

**Fig. 3 | TnpA^S911R-IR100 interactions. a** Interaction between IR100 and TnpA. Domains interacting with DNA are shown as space filling models. tr-LN: linker domain of trans subunit, tr-RNH: RNase H-like domain of trans subunit and tr-SCF: fragment of scaffold domain of trans subunit. **b**–**f** Details of DNA interaction with specific domains: DBD1 (**b**), DBD2 (**c**), DBD3 (**d**), DBD4 (**e**), and interactions with trans subunit (**f**). Base-specific interactions are shown by dotted lines. Residues and bases involved in base-specific interactions are labeled in bold.

density, in which DD was fitted as a rigid body, suggests that the interaction between DD is unchanged (Figs. 1d, 2b).

The 30 residues long C-termini interlock the protomers by docking conserved residues onto the surface of the adjacent protomer having a total interaction surface per protomer of ~1500 Å² (Supplementary Fig. 9b, Supplementary Table 4). In IR-bound complexes, the dimer is further stabilized through interactions with DNA, which also stabilizes C-termini-mediated dimerization nearly doubling the interaction surface (see below, Supplementary Fig. 9b, Supplementary Table 4).

In TnpA^S911R-IR100 complex, DNA remained base-paired throughout its length, but the outer flanking segment bent sharply by ~72° at the trans-DNA interaction site next to the DNA cleavage site (Figs. 2a–d, 3a, f). The bending site corresponds to highly conserved box A of the recognition sequence (Fig. 4b) and is likely generated primarily due to the electrostatic interactions between the outer flanking segment and DBD3 (Fig. 2f, g). This is evidently supported by the reduction in the substrate bending angle from 72° for IR100 to 54° for IR48, in which the interaction between the outer flanking segment (5 bps) and DBD3 was reduced (Fig. 2f, Supplementary Movie 2). The DNA binding and bending patterns are consistent with the DNA footprint analysis showing extended protection of the outer flanking segment together with hypersensitive sites around the TnpA cleavage sites in PEC[29].

In the structure of the strand transfer-like complex, TnpA^S911R-IR71st, in which the transferred strand of the IR end is disconnected from the donor and joined to the target DNA (Supplementary Fig. 3a), the outer flanking segment is rotated by ~120° around G0, separating ^C1^C_5 from ^C-1^C_3 by 17 Å (Fig. 2g, Supplementary Movie 2). This repositioning of the outer flanking segment suggests that stress release upon scissile bond cleavage is necessary to avoid clashes between the transposon ends and the target DNA in the active site, enabling the target DNA to approach the attacking 3'-OH group of the transposon end for the strand transfer reaction. The bending of outer flanking segments is also needed to provide available space for and avoid clashes between flanking segments of the donor and target DNA that should simultaneously bind to the active site (see below). The role of DNA bending at the ends of the transposon is mechanistically distinct from that generally evoked for the target bending, which is to prevent the strand-transfer reaction from reversing by moving the product 3'-OH group sufficiently far away from the scissile phosphate[31,33,34].

The design of the IR71st substrate does not allow complete annealing of the 5 nucleotides long single-stranded target DNA segments, and it, therefore, does not assemble into a canonical strand transfer product[36] in which 5 bps from the target remain base-paired after staggered strand transfer of the transposon ends. The DNA branch corresponding to the target is mainly disordered starting from the IR 3'-end cytosine on the transferred strand; however, a low-

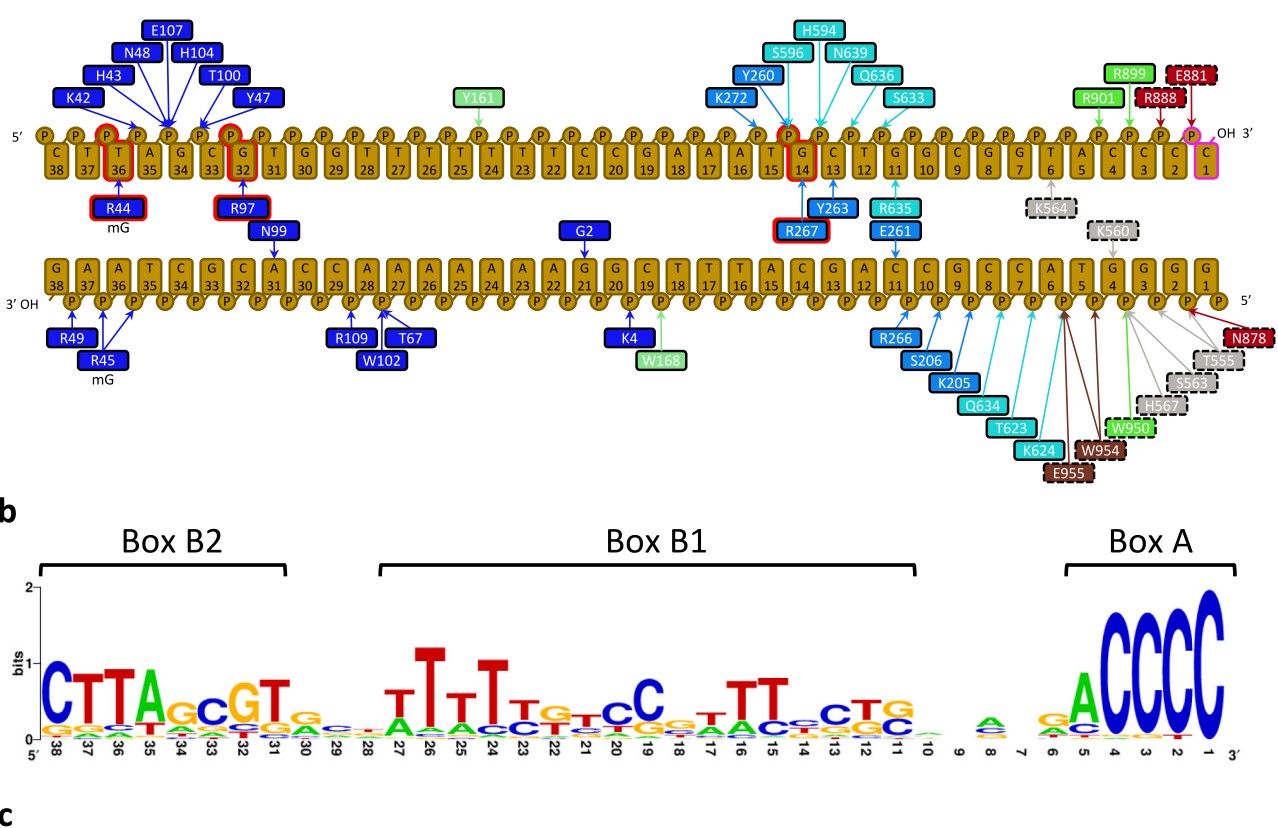

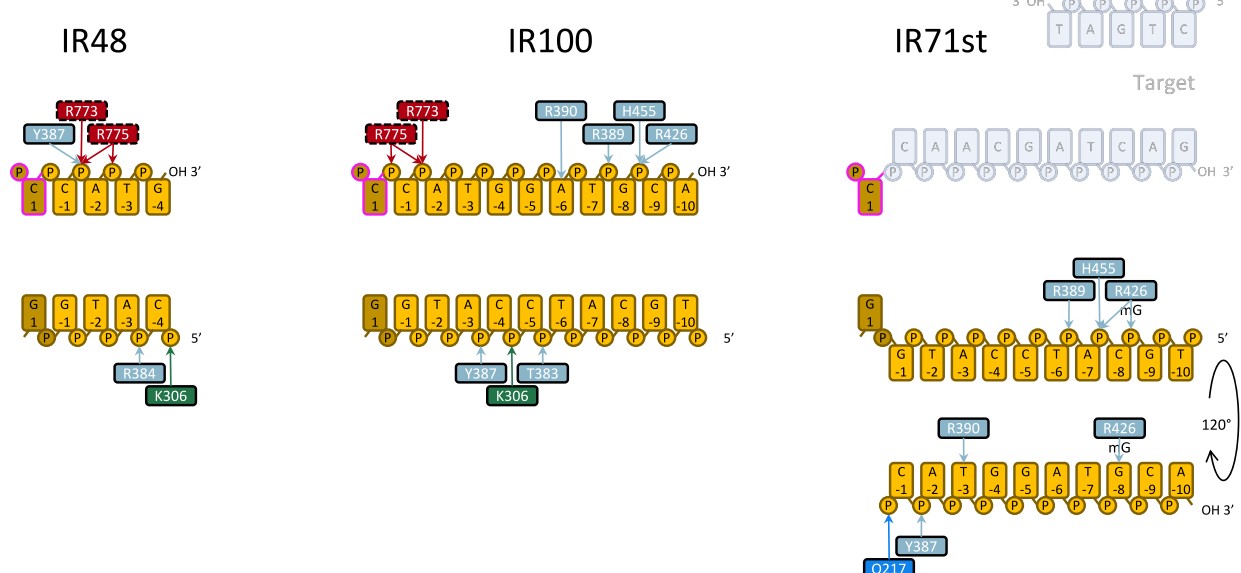

**Fig. 4 | Schematic representation of TnpA-DNA interactions and conservation of the recognition sequence. a** Interaction of TnpA with IR sequence. Interacting residues are color coded by TnpA domain as in Fig. 1. Residues and bases displaying high covariation in Tn3 family are shown with red borderline. **b** Conservation of the recognition sequence. Three conserved regions named Box A, Box B1, and Box B2 are indicated. **c** DBD3-outer flanking segment interactions for IR48, IR100, and IR71st constructs. In panels **a** and **c** residues are color coded by domain color as in Fig. 1d. Trans interactions are shown with dashed borderline around the corresponding box.

resolution density consistent with the DNA fragment was observed adjacent to the RNH domain in the map of TnpA$^{S911R}$-IR71st (Fig. 2d). This was attributed to the target-like branch of the substrate.

### Interaction with transposon ends

DNA substrates interact with four protein domains DBD1-4 in cis, and with RNH, LN, and SCF in trans (Fig. 3a). An extended 130 Å long positively charged DNA-binding surface (Supplementary Fig. 9d) creates over 50 polar contacts with a 47 bp long cis-DNA of the IR100 substrate and buries an area between 2700 and 3400 Å² (Figs. 2, 3, Supplementary Table 4) that includes the 38 bp IR sequence and a 9 bp outer flanking segment from the donor locus. The structural differences between the three TnpA-DNA complexes were primarily confined to the differences in the conformation of the outer flanking segment and its interactions with DBD3 (Fig. 2f, g, Supplementary Movie 2).

DBD3 binds the outer flanking sequences non-specifically through interactions with the DNA backbone (Fig. 3d) at positions −1 to −8. The lack of specificity of these interactions permits DNA binding to DBD3 at different positions and orientations (Fig. 2f, g, Supplementary Movie 2).

In contrast, DBD1, 2, and 4 interact with the IR recognition motifs in a sequence-specific manner (Figs. 3b, 4a). DBD1 is a key determinant of the specificity. Unexpectedly, it shares fold similarity and DNA-binding surface with BEN domains, a class of DBD found in a variety of transcription factors involved in chromatin silencing and gene repression in eukaryotes[37] (Supplementary Fig. 10a). DBD1 interacts with the conserved DNA sequence of box B2 of the IR and forms more than 20 polar interactions spread over a 50 Å long contact surface (Figs. 3b, 4a). Base-specific interactions occur within both the minor and major grooves between positions 21 and 36 (Fig. 2b).

One DNA helical turn down from the scissile bond (bps 6-15), DBD2 and DBD4 are docked into a major groove segment of the IR (Fig. 3c, e). DNA sequence recognition is mediated by a short α-helix (residues 261-267) on DBD2 (Fig. 3c) and residue R635 from DBD4 (Fig. 2e) and occurs with nucleotides C11, G11, C13 and G14 at the beginning of box B1, exhibiting a low level of conservation (Fig. 3a). DBD4 is structurally homologous and shares a DNA-binding mode with the N-terminal domain of Tn5 transposase[30] (RMSD 2.7 Å over 53 residues) despite only 9% sequence identity (Supplementary Fig. 10b).

TnpA trans-DNA interactions are mediated by LN, SCF, and RNH domains with bps between positions 1 and 6 and accounts for the contact surface of between 740 and 860 Å² (Supplementary Table 4). The interaction involves a highly conserved box A, viz. 5′GGGGT (Figs. 3f, 4a, b). Base-specific interactions with LN domain K560-G4 and K564-T6 contribute to DNA sequence recognition.

Consistent with the high specificity of TnpAs for their respective IR[10,28], sequence-recognizing residues and corresponding nucleotides, with the exception of R44-T36, display modest or no conservation (Supplementary Fig. 2), while the only two pairs, R97-G32 and R267-G14, display high covariance between amino acids and nucleotides (Fig. 4a). Therefore, conservation of the transposon recognition sequence[10] likely reflects the geometric constraints required for matching the DNA backbone to the extended DNA-binding surface of TnpA and IR recognition through an indirect read-out mechanism[38]. The DNA bend between the recognition sequence and the outer flanking segment occurs at the highly conserved sequence of box A at the very end of the transposon. Sequence conservation likely reflects the requirement for and mechanistic importance of outer flanking segment bending[39].

### Conformational changes

Upon binding to IR ends, the protein module upstream of DBD4, with the exception of DD, translates and rotates by ~50° as a rigid body, resulting in a shift of DBD1 by 40 Å (Supplementary Movie 3). These large-scale conformational changes are reminiscent to those observed

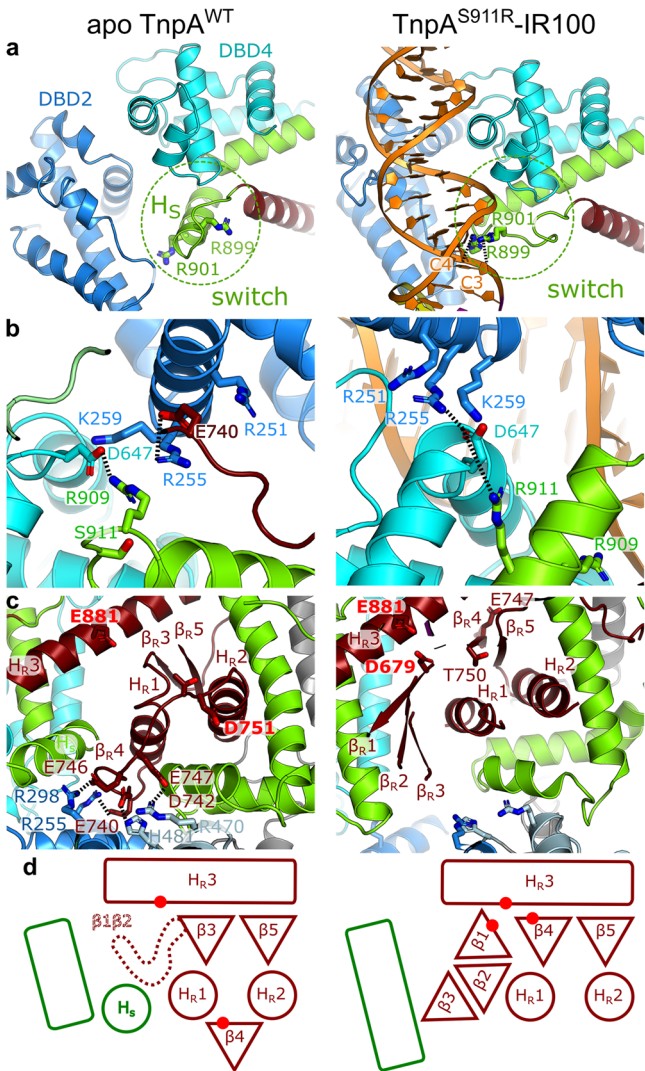

**Fig. 5 | Details of conformational changes between apo and TnpA$^{S911R}$-IR100 states. a** Rearrangement of DBD2 relative to DBD4 and refolding of switch helix $H_S$. The loop connecting $H_R3$ with $H_S$ caries two positively charged residues R899 and R901 that bind to cis-DNA. **b** Ionic lock, and salt bridge R911-D647 stabilizing the IR-bound conformation. **c** Metamorphic behavior of RNH domain that changes fold when switches from apo to IR-bound conformation. The catalytic residues are labeled in red. **d** Schematic cartoon showing rearrangement of the fold of structural elements in the catalytic site that includes straightening of the switch helix, repositioning of strands $\beta_R3$ and $\beta_R4$ and folding of strands $\beta_R1$ and $\beta_R2$. The rectangles and circles correspond to α-helices, and triangles to β-strands. Red dots indicate the positions of the catalytic residues.

in Transib transposase[31], despite the lack of primary sequence or structural similarities between the transposases. Conformational changes are a prerequisite for the tight binding of IR and formation of PEC and strand transfer-like complex. They render the surface of DBD3, otherwise occluded by the linker domain, accessible for DNA binding (Supplementary Movie 3), and rearrange DBD2 relative to DBD4 to form the DNA-binding site (Fig. 5a). The conformational transition also creates a 30 Å opening between the bodies of the protomers and DD, which is absent in the apo state (Figs. 1d, 2b, Supplementary Movies 1, 3).

Upon apo-to-IR-bound complex transition, several salt bridges present in a cavity proximal to S911 and connecting different domains are disrupted (Fig. 5b). The S911R mutation introduces an additional positive charge into the cavity, which likely destabilizes the electrostatic interactions and facilitates conformational transition. This

hyperactive mutation further stabilizes the IR-bound conformation by forming an R911-D647 salt bridge (Fig. 5b). Mapping other character-ized hyperactive and target immunity-deficient mutations[28] on the TnpA structure showed that most of them destabilize interactions between the domains found in the apo but not in the IR-bound con-formation (Supplementary Fig. 11), consistent with their higher pro-pensity than TnpA[WT] to form the PEC and to cleave the transposon ends in vitro[29]. This suggests that the apo-to-IR-bound conformational transition controls both the activity and target immunity of the transposase.

The apo-to-IR-bound conformational transition involves a pecu-liar conformational change in the conserved region of SCF. A helix (residues 897-908), further referred to as the switch helix, $H_S$, located downstream of the RNH domain, changes its fold. The N-terminal turn of $H_S$ (residues 897–901) unfolds in a loop, while the rest refolds into an extension of a long scaffold helix. This local refolding reorients R899 and R901, buried in the apo state, toward the protein surface where they interact with the cis-DNA backbone at positions C4 and C3, respectively (Fig. 5a, Supplementary Movie 3).

The insertion domain that interrupts the RNase H fold and the C-terminally located $H_S$, which together constitute the SCF, are the best-conserved regions of Tn3-family TnpAs, along with the RNH domain (Supplementary Fig. 2), thereby implying a functional role of SCF in transposition process. Intriguingly, an α-helix equivalent to $H_S$ has been observed in all currently available structures of DDE/D transposases, irrespective of their transposition mechanisms, and positions of $H_S$ homologs are structurally conserved relative to the respective RNH domains (Supplementary Fig. 5) despite the differ-ences in their lengths and sequences. Furthermore, the loop con-necting RNH and $H_S$-like helix carries two positively charged residues, structurally homologous to R899 and R901 in TnpA, which interact with the transposon end in cis, with the exception of the Mos1 and MuA transposases (Supplementary Fig. 5). These observations suggest that the loop and $H_S$-like helix are structurally and possibly functionally conserved elements in DDE/D transposases.

## Metamorphic behavior of RNase H-like domain

The conserved fold of the RNase H superfamily, three α-helices ($H_R$1-3) flanking a five-stranded β-sheet ($\beta_R$1–5)[19], is consistent with TnpA secondary structure prediction[28]. However, the RNH domain is among the least ordered parts of the complex and displays an unusual metamorphic behavior (Fig. 5c, d). In the apo conformation, adjacent α-helices $H_R$1 and $H_R$2 sandwich a short 2-stranded β-sheet ($\beta_R$3, $\beta_R$5) with catalytic $H_R$3, whereas the predicted $\beta_R$4 is positioned on the opposite side of the α-helical pair, where it is stabilized by forming salt bridges with DBD2 and DBD3 (Fig. 5c, Supplementary Fig. 12a). The helical scaffold is tightly wrapped around the RNH, with $H_S$ precluding the assembly of the 5-stranded β-sheet (Fig. 5c). Consequently, the densities of $\beta_R$1 and $\beta_R$2 were not observed in the apo conformation. In such a non-canonical conformation, RNH was partially disassembled whereas the DDE active site was completely disorganized (Fig. 5c, d).

In IR-bound complexes, refolding of $H_S$ together with rearrange-ment of the scaffold, allows for the folding of the RNH β-sheet, and a low-resolution density consistent with an extended β-sheet was observed in the 3D maps. The β-sheet was modeled using AlphaFold2[40] (Fig. 5d, Supplementary Fig. 12b, c). However, the densities of many loops connecting the strands were missing (Supplementary Fig. 12b), and therefore these loops were not modeled.

The DDE catalytic triad consists of residues D679, D751, and E881 (Supplementary Fig. 2). Among them, E881, which sits on $H_R$3, has a well resolved density and is exposed to the scissile bond of C1 (Fig. 2e) on the transferred strand (Figs. 2f, 4d), while D679 has a low-resolution density and D751 has no detectable density (the density stops at resi-due T750) in all three structures of TnpA-DNA complexes. This indi-cates that these residues are highly mobile, and their mobility is

independent of the presence of divalent ions (Supplementary Table 1). This observation is consistent with relatively low in vitro TnpA activity[29]. Interestingly, both catalytic aspartates are closely positioned in the model predicted by AlphFold2 (Supplementary Fig. 12c). This may indicate that their mobility is reduced in the fully catalytically active complex.

Protein metamorphism has been described as a regulatory strat-egy in several proteins[41]; however, to the best of our knowledge, it has not been observed for RNase H domains, and thus may represent a Tn3 family specific regulatory mechanism to control the transposase activity.

Comparison of TnpA with other structurally characterized trans-posases revealed structural homology between TnpA's DBD4 and DBD of the cut-and-paste Tn5 transposases[30] (Supplementary Table 3), along with similarity in relative positions of RNH and DBD4 domains between these two transposases (Supplementary Fig. 10c). However, in Tn5, the equivalent of SCF, encircling RNH domain, is missing, and metamorphic refolding was not observed[30,42]. This suggests a common evolutionary origin for TnpA and Tn5 transposases, in spite of their structurally and mechanistically distinct features.

The structures of apo and DNA-bound forms at different stages of transposition are also available for another transposase, Transib[31]. Unlike in TnpA, activation of Transib is not accompanied by meta-morphic refolding of the RNH domain or by a change in the fold of the switch helix; however, the loop connecting $H_R$3 with the $H_S$-like helix, does change the conformation upon activation in the strand transfer complex with simultaneous rotation of the $H_R$3 helix, leading to the assembly of the catalytic site (Supplementary Fig. 13). This suggests that the loop preceding the switch helix may function as a motif that recognizes the binding of cis-DNA and activates RNH domains for DNA cleavage in trans in the other transposases.

## TnpA[WT]-DNA complex and model of transpososome assembly

The structures of TnpA in the apo state and paired with transposon ends are dimeric. This observation contradicts the previously pro-posed model in which active transpososome assembles from TnpA monomers[29]. However, consistent with previous biochemical data[29], the structural signatures of TnpA[S911R]-IR71st suggested that it repre-sents a transpososome-like complex without the target DNA in the active site. A distance of ~30 Å between the symmetry-related scissile bonds is consistent with the 5 bp staggered insertion of the transposon ends into the target DNA[10]. The opening between the dimerization domains is sufficiently large to accommodate double-stranded DNA, and its surface is positively charged and highly conserved compared to the rest of the TnpA surface (Supplementary Fig. 9d, e). To further support these conclusions, a putative target DNA was modeled into the TnpA[S911R]-IR71st complex (Supplementary Fig. 14), such that double-stranded DNA threads through the opening and fits in the density assigned to the target-like DNA branch (Fig. 2d). The model of the target DNA does not clash with TnpA or outer flanking segments, and it is strongly bent as is commonly observed in other transposases[31,33,34].

Despite being mobile, the RNase H-like domain was assembled and appeared to be correctly positioned to cut the transposon DNA. The density of the target DNA branch next to the RNH domain indi-cates that the surface of the RNH domain has an affinity for DNA. The target DNA bound within the active site may in turn stabilize the RNH domain in its active conformation.

The double dimerization interface that closes the TnpA dimer is stabilized in the IR-bound complexes. The position of DD is well-defined relative to TnpA dimer, whereas on the other side the dimer-ization interactions mediated by C-terminal extensions are stabilized by the interactions with cis-DNA (Supplementary Fig. 9b, Supple-mentary Table 4), and the protomers are further cross-linked by the bound transposon ends. The target DNA is completely enclosed by the TnpA dimerization interfaces observed in both apo and IR-bound

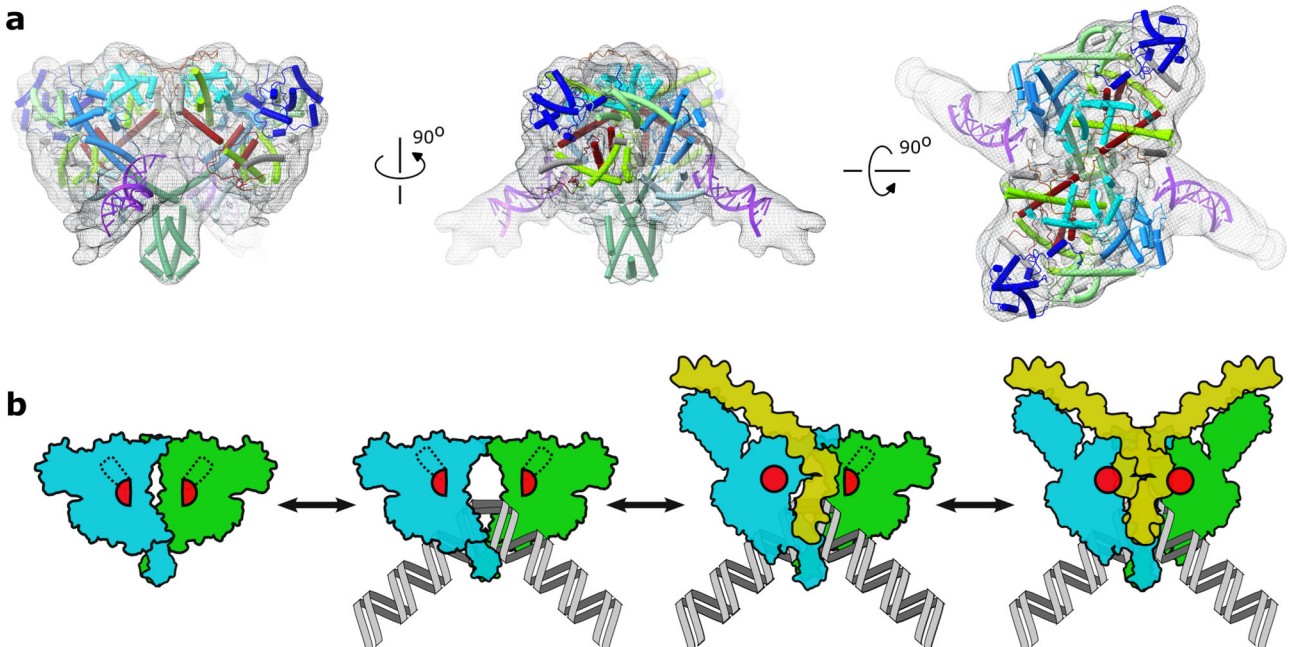

**Fig. 6 | Structure of non-activated apo-like state with bound DNA and proposed mechanism of transpososome assembly. a** Low-resolution cryo-EM map and fitted structure of TnpA^WT with bound IR100. TnpA^WT is shown in cartoon representation with domains color coded as in Fig. 1. The bound DNA, shown in purple, has position similar to outer flanking segment in TnpA^S911R-IR100 complex. In this conformation protomers have moved apart creating an opening for potential target DNA binding while conformation of DNA-binding domain corresponds to the inactive apo state. **b** Cartoon of the proposed mechanism for transpososome assembly. The RNase H-like domain is shown by a red semi-circle in an inactive apo state and by a complete circle in an active state. The target DNA is shown in gray.

complexes, thereby raising the question on how target DNA reaches its binding site. Although we cannot formally exclude the possibility that DNA could be threaded through the protein from a pre-existing double-strand break, it seems most likely that the TnpA dimer assembles onto the target prior to PEC formation.

The TnpA^WT dataset was collected in the presence of IR100 substrate. Even though the majority of particles were found in the DNA-free state, a smaller fraction of particles revealed the low-resolution reconstruction of TnpA^WT-IR100 complex (Fig. 6a, Supplementary Fig. 4b, c). Unexpectedly, the structure of TnpA^WT-IR100, rather than corresponding to the biochemically observed single-end complex[29] in which one transposon end is specifically bound to TnpA, has ends of the straight DNA fragments bound to DBD3 in the position occupied by outer flanking segments in the IR-bound complexes (Fig. 6a). In the TnpA^WT-IR100 complex, the protomers are moved apart, creating an opening that can accommodate the target DNA, yet each protomer is found in an apo-like conformation that cannot bind transposon ends. The structure of the TnpA^WT-IR100 complex demonstrated that the TnpA dimer is flexible and does not need to be fully activated to create an opening for target DNA binding. The single-end complex conformation was not unambiguously resolved, although 2D class averages with features expected for single-end complexes were observed in the TnpA^WT-IR100 dataset (Supplementary Fig. 4b, red box), indicating that a single-end complex might be present in the ensemble, albeit at a very low occupancy. This is consistent with previous biochemical and AFM-based single-particle spectrometry analyses, suggesting that the single-end complex is a short-lived transient intermediate of the transpososome assembly[43].

Taken together, our structural data suggest a plausible Tn3-family specific transposon assembly and activation mechanism, as schematically depicted in Fig. 6b. First, the target DNA binds, wherein the binding might be enabled by a spontaneous transient disruption of the dimerization interface mediated by C-terminal fragments or upon interaction with the target DNA, which might require to be in a specific conformation to become a permissive target. The binding of the target

DNA is followed by sequential binding of the transposon ends associated with conformational changes in TnpA protomers. They cause the switch helix of the corresponding cis protomer to refold, inducing the folding of the RNH domain of the same protomer. This results in the formation of a single end complex[29]. The binding of the second transposon end follows the same sequence of conformational changes in the second protomer, which concludes the assembly of the active transpososome.

Although this model is the most consistent with structural data, one cannot exclude the possibility that the binding of a single transposon end results in an asymmetric TnpA complex that has one of the dimerization interfaces disrupted, thereby facilitating the binding of target DNA.

The proposal that TnpA binds to the target DNA first suggests a plausible model for target immunity. The absence of DNA in the proposed target DNA binding site suggests that target DNA must have a specific conformation for binding or that the target DNA binding is the rate-limiting step of transpososome assembly. TnpA forms the active PEC on a fully assembled TnpA-target DNA complex in which the target DNA is enclosed and adequately positioned within the TnpA dimer (Fig. 6b). As described above, the formation of this complex requires that TnpA opens and closes around the target DNA, which may represent a slow step in the assembly process. Suppose the TnpA-target DNA assembles in the vicinity of a transposon. In that case, the interaction of TnpA with the transposon ends, prior to completion of target DNA binding, may lead to TnpA dissociation from or arrest of TnpA binding to the target DNA thus preventing the assembly of transpososome. Whenever TnpA associates with target on DNA regions remote from the transposon, the interference of TnpA-target DNA binding with transposon ends is reduced allowing for complete transpososome assembly.

## Discussion

Models for paste-and-copy replicative transposition mediated by Tn3-family transposons and other bacterial elements were among the first

to be proposed in the literature[36] and are presented as a classical mechanism of transposition in textbooks. This mechanism has been described in greater molecular detail for bacteriophage Mu, which uses replicative transposition to multiply its genome during lytic development[44]. However, the relevance of the Mu paradigm for non-viral elements, such as Tn3-family transposons, is questionable. In particular, Mu transposition is mediated by two main proteins: transposase MuA and ATP-dependent target-binding protein MuB, which is also involved in transposase activation and target immunity. The Mu transpososome is an oligomeric complex in which the core transposase consists of a tetramer of MuA[33]. MuA also stimulates ATP hydrolysis by MuB, promoting its dissociation from adjacent target DNA regions and making them immune to transposition. In the Tn3-family, TnpA is the only transposon-specific protein involved in both transposition and target immunity. The active form of TnpA appears to be a dimer, and no apparent nucleotide binding site was found in its structure and no ATP hydrolysis activity has been detected thus far. Hence, as shown here for Tn4430, the molecular architecture of the Tn3-family transpososome is different from Mu, and the mechanisms that control its assembly and activity are also likely to differ from those described for Mu[33].

Unlike cut-and-paste transposition that excise the transposon from the donor molecule by the formation of double-strand breaks at both ends, or the copy-out-paste-in mechanism during which replication generates a circular copy of the transposon prior to integration into a new locus[16,20,21], initiation of paste-and-copy transposition is a one-step process. It requires the assembly of an elaborate transposition complex in which two distant regions of the genome the donor and target, are brought together to catalyze single-strand DNA cleavage and joining reactions between the transposon ends and the target DNA (Fig. 1a)[10,16,44]. These reactions must be highly concerted and regulated because incomplete or abortive transposition can damage both the donor and target molecule, thus compromising the survival of the transposon.

The cryo-EM structures reported here reveal that the assembly of active Tn4430 transpososomes is controlled at multiple levels. Unusual folding of the RNH domain in the apo state likely ensures that the target DNA is not cleaved before the transposon ends bind. Thus, the refolding of the RNH domain into the active conformation is dependent on the binding of the transposon end, which refolds the switch helix, allowing folding of the RNH domain into an active conformation. Interestingly, the structural module associating the switch-like helix with the RNH domain appears to be conserved among DDE/D transposases, suggesting that such a coupling between transposon end binding and catalytic activation may represent a more general regulatory mechanism to control transposition, even with the less dramatic extension of conformational rearrangements. In the absence of target DNA but with bound transposon ends, the RNH domain assembles in an active conformation but remains very dynamic and has low activity, which may prevent it from cleaving the transferred strand in the absence of target DNA.

Our conclusions regarding the activated form of TnpA were based on the structures of the hyperactive mutant S911R. However, structural features of the mutant, including the conformation of the switch helix conserved in other transposases, the fold of the RNH domain, and preserved catalytic activity[29], suggest that the mutation does not create bias in the protein conformation. Moreover, 2D class averages of apo TnpA$^{S911R}$ indicated that in apo state its conformation is similar to that of TnpA$^{IWT\ 43}$.

Being coupled to DNA replication, paste-and-copy transposition is likely one of the most powerful mechanisms to promote the dispersal of foreign genes and to bring about specific DNA rearrangements, such as deletions, inversions, and replicon fusions, which have been shown to play a crucial role in bacterial genome evolution, notably by reassortment of multidrug-resistant plasmids in response to antibiotic pressure[45,46]. The resolving structure of a completely assembled TnpA transpososome should be the next important step toward understanding the transposition mechanism in atomic details.

## Methods

### DNA substrates

IR substrates were generated by annealing specific oligonucleotides (Supplementary Table 5, Supplementary Fig. 3a) at 95 °C for 10 min, followed by cooling to room temperature.

### Protein production, purification, and characterization

Tn4430 TnpA$^{WT}$ and TnpA$^{S911R}$ mutant were fused to a cMyc-His$_6$ epitope tag at the C-terminus and expressed in *E. coli* TOP10 cells under the control of the pAra promoter[29]. Cells were grown at 37 °C in TB media containing tetracycline (12.5 µg/ml) till OD600 reached 0.7–0.8; the temperature was then dropped to 18 °C. To induce the cellular chaperones, benzyl alcohol (0.1%) was added, and the cells were grown for 2 h before induction with L-arabinose (0.04%). After 3–4 h the medium was topped with L-arabinose (0.12%) and cells were grown overnight. The following day, the bacteria were centrifuged at 4 °C (7000 × *g*, 45 min), the pellet from 500 ml of bacterial culture (-5 g) was re-suspended in 20–30 ml of buffer A [50 mM HEPES (pH 7.9), 1 M NaCl, 10% glycerol, and 20 mM imidazole] supplemented with cOmplete EDTA-free inhibitor cocktail tablet (Sigma-Aldrich), and flash frozen in liquid nitrogen. The thawed cell suspensions were supplemented with 0.25 mg/ml lysozyme (Sigma-Aldrich), 0.1% Triton-X (Sigma-Aldrich), MgCl$_2$ (10 mM), and DNase I (Sigma-Aldrich). The mixture was diluted to a final volume of 20 ml with buffer A and incubated for 1 h at 4 °C on a rotator. After sonication, the lysate was cleared by centrifugation (18,000 × *g*, 45 min), filtered through a 0.45 µm filter, and supplemented with 5 mM ATP and 4 mM MgCl$_2$ before loading on a 5 ml HisTrap column (Amersham) pre-equilibrated in buffer A. Next, the bound material was washed with two column volumes (CV) of buffer A containing 0.1% Triton-X and two CV of buffer A containing 5 mM ATP and MgCl$_2$ interspersed by washes with buffer A. TnpA was eluted with a 20–500 mM linear gradient of imidazole in buffer A over 16 column volumes. Pooled fractions were concentrated by ultrafiltration to -350 µl (Amicon Ultra-15, 100 kDa MWCO) and applied to a Superose 6 Increase 10/300 GL column equilibrated in 50 mM HEPES (pH 7.9), 200 mM NaCl, and 100 mM L-Arg HCl. Fractions containing pure TnpA were pooled and mixed with a 4- to 10-fold molar excess of DNA substrates. The protein and DNA concentrations during the complexation of TnpA$^{WT}$ with IR100 were 0.5 and 2.5 µM, while the complexation of TnpA$^{S911R}$ with DNA was performed at concentrations of 2.5 and 10–25 µM, respectively. Owing to its low stability, TnpA$^{WT}$ was complexed with IR100 for 1 h at 4 °C after which it was plunge-frozen. TnpA$^{S911R}$ was incubated with IR substrates overnight, concentrated to -350 µl (Amicon Ultra-15, 100 kDa MWCO) and subjected to size-exclusion chromatography to remove unbound DNA using Superose 6 10/300 GL (GE Healthcare) column equilibrated in 50 mM HEPES (pH 7.5), 100 mM NaCl, and 30 mM L-Arg HCl. Freshly purified TnpA-DNA complexes were directly used for the preparation of cryo-EM grids.

The homogeneity and oligomeric state of the apo and complex forms were assessed by mass photometry on a Refeyn OneMP instrument (Refeyn Ltd.), which was calibrated using an unstained native protein ladder (NativeMark™ Unstained Protein Standard A, Thermo Fisher Scientific Inc.). Measurements were performed at concentrations of 0.1–0.2 mg/ml using AcquireMP 2.2.0 software and were analyzed using the DiscoverMP 2.2.0 package (Supplementary Fig. 3d).

### Preparation of cryo-EM grids

Quantifoil or C-flat holey carbon grids (R2/1, 300 mesh) were glow-discharged using ELMO system (Cordouan) at 0.3–0.35 mBar and current of 10–15 mA for 60 s. To prepare graphene-oxide-coated grids,

the aqueous dispersion of graphene oxide (GOgraphene; William Blythe Ltd) was diluted in double-distilled water (ddH$_2$O) to a final concentration of 1.3 mg/ml, followed by sonication in Elmasonic S 30 (H) for 120 s in a cold room and spun down at 300 g for ~2 min. C-flat holey carbon grids (R2/1, 300 mesh) were glow-discharged as described above, and 4 µl of GO solution was applied to the grids, followed by one minute incubation; subsequently, the GO solution was removed by blotting briefly with Whatman No.1 filter paper and washed by applying 20 µl ddH$_2$O onto the graphene-oxide-coated side twice and once on the back side of the grid with blotting steps in between.

A volume of 5 µL of TnpA$^{WT}$-IR100 mix (protein concentration of 0.06 mg/ml) was applied to a glow-discharged Quantifoil holey carbon grid (R2/1, 300 mesh), blotted from back side for 3 s at 70–90% relative humidity and plunge-frozen in liquid ethane using a Cryoplunge 3 System (Gatan). TnpA$^{S911R}$ complexes with DNA substrates (5 µl) were applied to GO-coated C-flat holey carbon grids (R2/1, 300 mesh) at protein concentration of 0.18 mg/ml, blotted and plunge-frozen as described above.

## EM data acquisition
The TnpA$^{WT}$ was imaged at the CM01 beamline at ESRF[47] using EPU v1.11 software for automated data acquisition on a Titan Krios cryo-electron microscope (Thermo Fisher Scientific) operated at 300 kV equipped with a Quantum LS electron energy filter (Gatan). Image stacks were recorded with a K2 Summit (Gatan) direct electron detector operating in counting mode at a recording rate of 4 raw frames per second. The microscope magnification of 130,000X (corresponding to a calibrated sampling of 1.067 Å per pixel) was used. The total dose was 50 electrons per Å$^2$ with a total exposure time of 10 s, yielding 40 frames per stack. A total of 3724 image stacks were collected with a defocus range of 0.6–5.3 µm (see Supplementary Table 1 for details).

Micrographs of TnpA$^{S911R}$ complexes with DNA were collected at 300 kV on a CRYO ARM 300 (JEOL) electron microscope at a nominal magnification of 60,000 and corresponding pixel size of ~0.76 Å. The images were recorded using a K3 detector (Gatan) operating in correlative-double sampling (CDS) mode. The microscope illumination conditions were set to spot size 6, alpha 1, and the diameters of the condenser and objective apertures were 100 and 150 µm, respectively. The energy filter slit was centered on the zero-loss peak with a slit width of 20 eV. Coma-corrected data acquisition[48] was used to acquire between 6 and 25 micrographs per stage position using SerialEM v3.0.8[49]. Each micrograph was recorded as a movie of 59 or 60 frames over a 3 s exposure time and at a dose rate of 11 e⁻pixel⁻¹s⁻¹ (corresponding to a dose rate per frame of 0.6 e⁻Å⁻²) and a total exposure dose of ~60 e⁻Å⁻² (see Supplementary Table 1 for details).

## Image processing
Initial data processing was performed on-the-fly using RELION_IT[50]. Dose-fractionated movies were subjected to motion correction and dose weighting using MotionCorr2[51]. The dose-weighted aligned images were used for CTF estimation using the CTFFIND-4[52]. An in-house script was used to plot the calculated parameters, visualize the results, and select micrographs for further processing (Shkumatov et al; in preparation). The aligned and dose-weighted images were imported into cryoSPARC v3.1.0[53], and CTF was calculated using Patch CTF. Particle selection was performed using a blob or a template-based picker, followed by several rounds of 2D classification. An ab initio reconstruction and initial 3D refinement were performed using cryoSPARC. 2D classification of the TnpA$^{WT}$ dataset in cryoSPARC revealed four different populations of classes, including higher-order oligomers (Supplementary Fig. 4b, blue frame). The different conformations were further separated using ab initio model calculations and heterogeneous refinement. Separated subsets were independently

reconstructed by applying homogeneous and non-uniform refinements (Supplementary Fig. 4). For the processing of the TnpA$^{S911R}$ datasets, particles were imported into RELION 3.1[50]. The low-pass filtered to 60 Å initial model was used for 3D auto-refinement using C1 symmetry. This was followed by multiple rounds of 3D-refinement and 3D classification using either C1 or C2 symmetry, CTF refinement, and Bayesian polishing[54] (Supplementary Figs. 6–8). To improve the density corresponding to the N-terminal domain in the TnpA$^{S911R}$-DNA complexes, the signal for the monomer was subtracted, followed by multibody refinement using two rigid bodies[50] (Supplementary Fig. 7). Local resolution was estimated in RELION 3.1, with a B-factor from the post-processing job. The directional resolution of the final map was measured using a 3DFSC server[55].

## Model building and refinement
Initially, parts of the model were built automatically using PHENIX v1.19.1 map_to_model procedure[56]. This was followed by manual model building in COOT 0.9.5[57]. To build a poorly resolved β-strand of the RNase H-like domain in TnpA$^{S911R}$-DNA complexes, the entire structure of TnpA was predicted using AlphaFold2[40] (Supplementary Fig. 12c), and the complete predicted RNH and scaffold domain was fitted into the density as a rigid body. Next, the regions in which density was absent were removed from the model. The dimerization domain (DD) was poorly resolved in TnpA$^{WT}$ apo map. Therefore, the domain was first built and refined in the TnpA$^{S911}$-IR100 map and then fitted into the TnpA$^{WT}$ apo map as a rigid body. The models were refined using PHEINIX v1.19.2 real_space_refine procedure[58] against maps filtered using the local filter procedure of RELION 3.1. Secondary structure, Ramachandran, and ADP restraints were applied during the refinement procedure that included 'global_minimization' and 'local_grid_search' strategies. ADP restrains were relaxed for the TnpA$^{S911R}$-IR71st complex during the last iteration. The models were validated using MolProbity[59]. Supplementary Table 1 presents the models and data statistics. The TnpA$^{WT}$-IR100 model was constructed by first fitting refined TnpA$^{S911R}$-IR100 in the low-resolution density map followed by real space refinement of the model in COOT with applied ProSMART restraints using the initial TnpA$^{S911R}$-IR100 complex as a reference model. The resulting structure was not further refined because of the low resolution of the map.

## Visualization, sequence alignment, and structure analysis
Protein images were prepared using PyMol v2.4.2 and ChimeraX v1.2.4[60] programs. Sequence alignments were performed using Clustal Omega[61] and visualized using ESPript 3.0 server[62]. A phylogenetic tree was generated using the TnCentral database[63]. The fold similarity was analyzed using the DALI server[64] and protein–protein contacts were calculated using PISA[65].

## Reporting summary
Further information on research design is available in the Nature Research Reporting Summary linked to this article.

## Data availability
The cryo-EM density maps and atomic models generated in this study have been deposited in the PDB and EMDB database under accession codes: for TnpAWT 7QD8 and EMD-13910), for TnpAS911R-IR100 7QD4 and EMD-13906), for TnpAS911R-IR48 7QD5 and EMD-13908), and for TnpAS911R-IR71st 7QD6 and EMD-13909). The atomic models used in this study are available in the PDB database under accession code 4X0G, 1MUH, 6PQN, 6PR5, 6XGX, 6P5A, 6X67, 4D1Q, 6PQU, 5HOO, 4FCY, 6B40.

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

## Acknowledgements

We are indebted to Dr. Adam Schröfel and Dr. Marcus Fislage for their assistance with cryo-EM data collection, to Dr. Gipsi Lima-Mendez for her valuable assistance and expertise in phylogenetic and bioinformatics analyses, and Dr Patrice Soumillion for helpful discussions about the structures and their interpretation. We acknowledge the European Synchrotron Radiation Facility for providing the beam time for CM01, and we would like to thank M. Hons for assistance. This work benefited from access to the Netherlands Center for Electron Nanoscopy (NeCEN) at Leiden University with the assistance of Dr. C. Diebolder and Dr. J. Ortiz Espinoza. We thank Prof. S. Raunser and Dr. O. Hofnagel for providing access to the electron microscope at the Max Planck Institute for Molecular Physiology. We thank the VIB Tech Watch fund for facilitating access to the Refeyn Instrument. We would like to acknowledge the funding provided by Vlaams Instituut voor Biotechnologie, Fonds Wetenschappelijk Onderzoek (Grant Nos. G0H5916N, G054617N to R.G.E.), and Fonds Spéciaux de Recherche (UCLouvain) and Fonds National de la Recherche Scientifique (CDR grants J.0162.16 and J.0096.20 to B.F.H.).

## Author contributions

A.V.S., C.A.O., G.G., N.A., and B.F.H. designed the strategy and DNA substrates for the preparation of cryo-EM samples. A.V.S. prepared the cryo-EM grids, collected the EM data, and processed the cryo-EM data. R.G.E. built and refined the models. All authors analyzed the results. R.G.E. prepared the original manuscript draft. A.V.S., N.A., B.F.H., and R.G.E. reviewed and edited the manuscript. A.V.S., N.A., and R.G.E. prepared the figures. B.H. and R.G.E. conceived, managed, and supervised the project and acquired funding.

## Competing interests

The authors declare no competing interests.
