## [Peer Review File · Nature Communications]

Structural insight into Tn3 family transposition mechanismREVIEWER COMMENTS

Reviewer #1 (Remarks to the Author):

In this article Shkumatov et al describe four structures of a Tn3-family transposase in apo DNA bound forms. Mobile genetic elements of the Tn3 transposon family are widespread in bacteria and contribute to the emergence and dissemination of antibiotic resistance genes. Thus, it is important to understand their molecular mechanisms.

The current study provides a great step forward in elucidating the structural basis of Tn3 transposition and it reveals common and distinct features with other transposition mechanisms. The structures are very nice and informative, but further analysis and discussion is needed to clarify their exact meaning and relevance for Tn3 transposition and beyond.

Main points:

1. One of the most interesting insights of the present work relates to the conformational changes accompanying transposon DNA binding and activation. In fact, the structure of the DNA-free apo transposase is only known for very few elements, hampering our overall understanding of the earliest steps of transposition. Perhaps the most prominent example, where the apo-PEC transition has been visualized, is for the Transib transposon (Liu et al. Nature 2019). That study showed a similarly significant rearrangement of the transposase structure from a compact apo form to a DNA complex with extended "wings". Consistently, I personally expect that major conformational changes accompany DNA binding in most transposon systems, and unlike the authors predict, are not unique to Tn3 family elements. Differences in the extent and exact nature of the changes will certainly exist though. In this context, it would be valuable for the authors to compare the Tn3 and Transib structures and discuss differences but also possible parallels, with other transposition pathways.

2. While the cryoEM structures are intriguing and of overall high quality, the manuscript lacks functional validation of the structural findings. A good amount of published biochemical data already support certain structural features, but the functional significance of novel unexpected findings should be tested. For example, is the refolding of the switch helix needed for transposition? Is the surprising "fluidity" of the RNH fold relevant in the natural context (e.g. WT TnpA) and does it contribute to transposition efficiency?

3. The peculiar rearrangement of the internal folding of RNH domain itself is very surprising and additional functional validation would be helpful. Also, what are the specific features of the Tn4430 TnpA that underlie this structural fluidity? Are these features shared by (some) other DDE transposases?

4. The manuscript describes all structures in parallel, comparing selected aspects in four structures simultaneously. This makes the flow rather difficult to follow, and the reader easily gets confused about the different pieces of data. I had to go back and forth in the text and figures constantly to remind myself about the content and appearance of the different structures to grasp and appreciate the various descriptions and discussions. The description of the TnpA IR71st complex is particularly confusing: it is never specifically described really, and it is not clear if the authors consider this finally as an STC or PEC.

Smaller points:

- The strategy of using a hyperactive variant to capture a stable PEC structure is clever. However, I wonder if the mutation partly biases transposase conformation. An assessment of the S911R apo (or the WT PEC) conformation would be helpful to clarify this possibility.
- It would be valuable to analyze and discuss the broader relevance of the results within and beyond the Tn3 family. For example, the authors state that the DNA binding residues are not conserved in the Tn3 family. But are they conserved in Tn4430 copies? And how about the switch helix, is this

conserved in the Tn3 family? Are the specific features of the RNH domain that may underlie its plasticity conserved?

- The Figure legends should describe the illustrations more specifically. It is often difficult to guess what exactly is shown.

- When describing the TnpA-IR71st structure, it should be stated clearly that the target mimicking stem is poorly visible in the EM maps. I was confused about this until the Discussion.

- Lines 93-94: Are the domain folds truly novel? Does a DALI search with individual domains give no meaningful hits? Some domains certainly have known folds – please correct.

- Lines 235-237: I do not understand how the authors arrive to the conclusion that the target DNA plays a role in activation and target immunity. I did not find any data that would support this proposal. Please clarify or remove.

- Line 258: Why do the authors expect to see a SEC in the TnpAwt-IR100 complex? Please explain.

- Lines 276-286: I cannot follow the authors model for target immunity. Perhaps I am missing something, but for me it seems that target tethering would lead to “local hopping” rather than target immunity.

Reviewer #2 (Remarks to the Author):

This is a very interesting paper that describes for the first time a series of DNA transposase and transposase/DNA complex structures from the Tn3 family. One aspect that makes the result novel is that this is the first time that a full-length replicative DNA transposase has been captured both in the apo and transposon DNA end-bound forms. A second novel aspect is that the transposase is already a dimer in the apo form and remains as a dimer in the transposon end-bound forms (PEC) as well, without forming higher order assemblies as observed for other similar systems (eg. phage Mu transposase). Furthermore, as the authors point out, the architectures of the assemblies and the two independent dimerization interface suggest that it is very likely that the target DNA binds first and the transposon ends are subsequently captured by the transposase/target DNA complex. While this is only a suggestion given the lack of clear target DNA density, it is very plausible and consistent with the available data, providing substantial mechanistic insight into Tn3 transposition. Given the importance of mobile genetic elements, and Tn3 in particular, in the spread of antibiotic resistance, I believe the results should be of wide interest.

That said, the paper needs several major issues to be addressed before publication.

1. Throughout the text when describing the results, it is not clear which of the four solved structures authors are referring to. As there are three different PEC structures, a consistent and definitive notation should be applied and used whenever a particular structure is analyzed or discussed. When a particular feature is common between all three PEC structures, this could be noted as "in all three PEC structures".

2. The paper needs much better figures. The assemblies are complexes with many domains and protein/protein and protein/DNA interactions. The chosen representations just do not do the job, are quite dark, and in particular Fig1d, Fig5a,b are not sufficient and Fig4a not at all useful. I very strongly urge the authors to include a schematic representation such as Figure 2a in Montano et al. (2012) Nature 491, 413 to show the essence of the assemblies. I should also like to point out that having a 3D cartoon representation in the same orientation adjacent to the schematics (as in Montano et al.) gives an unambiguous perspective and provides clear understanding. The use of a reasonable amount of fog on the structure figures (e.g., Fig 2b,c Montano et al.) is very useful for depth cueing.

3. The authors write about and display the conformational morphs between apo and PEC states. However, I find a logical inconsistency as the model in Fig5c describes an apo -> target bound -> target+transposon end bound progression. Therefore, I do not understand the point of discussing

much about the apo -> PEC conformational transition as this does not appear to be really relevant, unless of course I'm missing something.

4. I do not understand why the DD dimerization domain was not docked into the apo density. This dimerization interface is highly hydrophobic, and I would think that this is primarily responsible for holding the apo dimer together. "Protein-protein dimerization interfaces are very similar between apo and PEC conformations" yet the apo structure has no coordinates for the DD. I believe authors should consider that the apo form opens up at CT, while DD holds together to capture the target DNA.

5. Please compute the sizes of both CT and DD dimerization interfaces.

6. If target DNA is indeed bound before the transposon ends are bound, authors should show how and why target DNA would not sterically conflict with the DNA flanking the transposon. This is because, when transposon DNA is bound, it has a flank and it has to go somewhere, and in fact two of the PEC structures indicate where that would be. This is a significant mechanistic point, as there are other known DNA transpososome structures where target DNA clearly binds in the transposon DNA flank binding site, unambiguously establishing the time sequence of the events. If Tn3 is different, transposon flank and target DNA must coexist even if transiently.

7. I believe authors should state clearly that the structural results presented and the resulting model is inconsistent with the model for the transpososome assembly pathway proposed in Nicolas et al. 2017. The inconsistency goes beyond the notion of the monomeric nature of apo form of the transposase.

Minor comments:

Line 80. "The scissile phosphate" needs a figure. While I understand that part of the active site is disordered in all of the PEC structures, a reader would probably like to see the relative position of the scissile phosphate to the two visible carboxylates.

Line 81. Please do not use "cross-reactivity". The common terminology in the field is trans activity, in which the protomer's active site processes the other end relative to where its site-specific DNA binding domain is bound.

Line 92. "Structurally unique insertion domain". I do not see the results of a Dali search run with the coordinates to establish uniqueness. This has to be done. Are the authors certain that there is no similarity with the Rag1, Hermes and P element insertion domains?

Line 94. Did a Dali search provide no homologs to DBD2 or DBD3?

Line 99. As in major comment 4, please dock the DD domain into the low resolution density or give a clear reason why this cannot be done given that its structure is known from the PEC structures.

Line 104. "unusual dimerization interface". Unusual in what sense?

Line 105. "important consequences for the transposition mechanism". What consequences in particular?

Line 108. "cis-DNA". This is insufficient. Please define this DNA better for the general reader not familiar with the terminology of the field.

Line 116. "reduction in the bending angle". Reduction relative to what?

Line 122. "broken transferred strand". Please explain why the transferred strand is broken. I think this

is because of how the IR71 substrate was assembled, but this might not be clear for the general reader. I also note that in Methods the complex assembly section (after line 335) is only cursory and not detailed enough as written for a general reader. The relative protein DNA concentration ratios are missing for the PEC complexes.

Line 125. "making transposition irreversible". Please expand to explain this to the uninitiated.

Line 126. "DBD3-OFS interaction" is labeled as DBD4 in the legend of Fig 2d.

Line 127. "The interaction is also..." Please rewrite this sentence, explain it better, and fix the punctuation.

Line 134. "DNA-binding domain surface with BEN domains". How was this similarity found? Was a Dali search run? What were the results?

Line 176. Given that it is the nature of the mutation (S911R) that is informative - not simply that it was mutated - it would be nice to see Extended Data Fig 8 modified to specify the mutations, not just indicate their positions.

Line 198. "D751" I see no evidence of D571 in any of the PEC coordinate files, so I'm not clear how statements about it can be made. Please clarify.

Line 200. "additional factors are required". This is mysterious, so please explain what additional factors could be involved.

Line 202. "unique regulatory mechanism". Please explain how this is a regulatory mechanism.

Line 215. "MuB". Please mention a main feature of MuB is that it is an ATPase and the key factor in mediating target immunity in Mu replicative transposition. The point is that Tn3-like elements appear to display target immunity without an ATPase.

Line 220. "unique and totally differ from those of Mu". This is imprecise. Please explain how it differs.

Line 273. "cis switch helix" This is not defined, so it is not intelligible. Although it sounds interesting, it needs a better explanation and perhaps a figure somewhere?

Line 279. "transposon is already present in the target". Is the presence of the entire transposon required for immunity?

Line 280. "transposon recognition motif". Please define this with the terminology already used in the paper.

Line 284. "Whichever the..." Please rewrite this sentence. I have no idea how transposition can be effective without all substrates present.

Line 400. "were predicted using AlphaFold2". I do not understand this. I assume that AlphaFold2 predicted the structure of the entire RNaseH-like catalytic domain, so why wasn't such a model placed in the PEC structures based on alignment between strands/helices that are visible in the density? Or was this tried but resulted in clashes? If so, what would be the explanation? I would like to know what sequence was input to AlphaFold2, was it the entire transposase, or only the RNaseH-like catalytic domain? What were the results, how does that compare with the experimental structure?

Table 1. Ramachandran plot. My hope is that not almost 100% of all residues are in the disallowed regions. Please fix.

The movies. While I like the movies, they should have legends describing what are we looking at. Also, can they please be retraced at higher resolutions? Please use the "viewport" command in pymol, with some suitable large numbers while keeping the correct aspect ratio.

Reviewer #3 (Remarks to the Author):

This manuscript describes the long-awaited and quite surprising structure of Tn3 transposase in complex with DNA substrates. Beyond its historical and medical significance, Tn3 transposase has a feature that could make it more useful than other transposases for, say, transposon library construction: transposition and target immunity functions within a single protein. Overall, I am very enthusiastic about this major advance. The models appear to be carefully built and refined, and the results are thought-provoking – particularly the idea that in this system, the target DNA may bind BEFORE the transposon ends, and that that might be a key feature in target immunity. However, I found the presentation rather confusing.

Substantive comments:

- 1) Table 1 has typos that slander the nice models: it says 98% of the residues are in the disallowed Ramachandran plot regions, rather than in the favored regions (as shown in the validation report and the fact the the model has nice secondary structure).
- 2) The beginning of the Results section points out that substrate IR71st included target branches, but the lack of the target DNA in the figures isn't explained until close to the end. I found this very confusing.
- 3) Is the target in the IR71st complexes still covalently attached? Two reasons for wondering if it is not are: (1) Mu transposase, if given a similar substrate, has trouble figuring out which branch was supposed to be target vs. flanking donor DNA, and removes some of the target branches as hairpins (see Mizuuchi ... Mizuuchi PNAS 2007) and (2) the IR71st model has nice low B factors up to and including the 3' end of the transferred strand, then the DNA just disappears – exactly what one would expect if the target had been accidentally cleaved off as in that 2007 paper.
- 4) p6 – idea that bending of the flanking DNA could contribute to irreversibility: while it may be tempting to draw the analogy to the role target DNA bending in irreversibility, I don't think that makes sense. The initial cleavage step is a very favorable hydrolysis reaction that is irreversible anyway (you'd need ATP and ligase to religate that nick). The strand transfer step is where irreversibility requires an explanation, because that is not "downhill" in terms of high-energy chemical bonds. I would guess that the reason for bending the flanking donor DNA is to prevent steric clashes with the target DNA.
- 5) Same argument against the logic of p6 lines 129-130: initial hydrolysis is expected to be irreversible no matter what. The key problem for this type of transposase at this step is probably to get the flanking donor DNA out of the way of the target, which has to access the same active site.
- 6) I think the authors are really "onto something" regarding the connection between Tn3 transposases's mysterious target immunity and the idea that it binds target 1st rather than last. However, the discussion of possible mechanisms on p. 13 is confusing. By "tethering" do the authors suggest that the transposase initially forms a sliding clamp on non-specific (target) DNA? What is the "transposon recognition motif" referred to? (the transposon ends or something else?) What exactly is meant by avidity? Cartoons may help. Could it be that sliding of the transposase complex along target DNA results in a non-productive complex (and perhaps dissociation) if it slides onto a transposon end, thus engaging the 1st couple of DBDs while still binding part of the same DNA molecule in the target-binding spot, but results in productive transposition if transposon ends that are on a separate DNA segment join the complex in trans?
- 7) p8, bottom and extended data Fig. 8: do hyperactivity and target immunity generally correlate? And are some of these mutations at protein-protein interfaces in the full complex?
- 9) Mu transposase also has a protein-protein interaction segment blocking "escape" of the target DNA, but Mu is known to bind target last. This might be worth discussing, especially because the dimerization domain seems to be disordered in the apo structure.

10) p. 6, top refers to a "box 1" in Fig 3 – is that "box A" in the figure?

Suggestions (these are not demands) for making the manuscript easier to understand:

1) if possible, redo many of the figures with better depth cueing (fading out in the back) and without shadows – for a structure this big, the shadows just add visual noise.

2) In the 1st paragraph of results, mention that the target DNA is only partially visible in the IR71st complex, and that there is a hint of target visible in some of the "apo" complexes – then just say it will be discussed later. That will keep the reader from wondering what happened to it.

3) In figures such as 1d, explain what the white blob is (electron density or space-filling outline?)

4) Cut down on the number of acronyms the reader has to memorize to follow the paper. Unless this journal is too strict about character limits, spell out things like "paired end complex," "single end complex" and "outer flanking segment." (the protein already has 10 domains, each with its own acronym).

5) The color key in Figure 1 says bright yellow is donor DNA – but that seems to be referred to as OFS in other places

6) I'm sorry but I've started at Figure 4a for a long time, and I still can't follow what is going where except for DBD1 (which should be labeled as such). Perhaps showing the models side-by-side rather than superimposed would help.

7) cross-reactivity is confusing because that term can also mean, say, enzyme 1 also works on enzyme 2's substrate. Isn't this usually referred to as "catalysis in trans"?

8) a figure comparing the RNaseH domains from these structures to some seen in previous DDE enzymes would be helpful.

9) Work with a native speaker of English to polish the writing.

We are thankful to the reviewers for their positive appraisal and thorough reading of the manuscript. We found the detailed remarks and comments very valuable and constructive, based on which we revised the manuscript aiming at introducing corrections accordingly.

The major corrections include:

- (1) Updated structure of TnpA^{WT} with fitted dimerization domain,
- (2) Figures of the main text were remade. Figure 1 was split into 2 figures Fig. 1 and Fig. 2
- (3) We improved the flow of the manuscript, particularly in the first half of the results section describing the overall architecture of TnpA and its complexes with DNA.
- (4) Movies were retraced at higher resolution and annotated.
- (5) Supplementary Figures with a comparison of TnpA to other transposases were added: Supplementary Figs. 5 and 13.
- (6) Supplementary figure 14 with a putative model of target DNA was added.

The modifications are highlighted in yellow in the revised manuscript. Below we provide point-by-point answers to the reviewers' comments.

Reviewer #1 (Remarks to the Author):

In this article Shkumatov et al describe four structures of a Tn3-family transposase in apo DNA bound forms. Mobile genetic elements of the Tn3 transposon family are widespread in bacteria and contribute to the emergence and dissemination of antibiotic resistance genes. Thus, it is important to understand their molecular mechanisms. The current study provides a great step forward in elucidating the structural basis of Tn3 transposition and it reveals common and distinct features with other transposition mechanisms. The structures are very nice and informative, but further analysis and discussion is needed to clarify their exact meaning and relevance for Tn3 transposition and beyond.

We are thankful to reviewer #1 for the positive feedback regarding the relevance and interest of our work, and for having encouraged us to deepen the analysis of our results by comparing TnpA to other transposases and integrases. We performed additional comparisons of the structure which we believe further strengthen our conclusions with an even more general scope.

Main points:

1. One of the most interesting insights of the present work relates to the conformational changes accompanying transposon DNA binding and activation. In fact, the structure of the DNA-free apo transposase is only known for very few elements, hampering our overall understanding of the earliest steps of transposition. Perhaps the most prominent example, where the apo-PEC transition has been visualized, is for the Transib transposon (Liu et al. Nature 2019). That study showed a similarly significant rearrangement of the transposase structure from a compact apo form to a DNA complex with extended "wings". Consistently, I

personally expect that major conformational changes accompany DNA binding in most transposon systems, and unlike the authors predict, are not unique to Tn3 family elements. Differences in the extent and exact nature of the changes will certainly exist though. In this context, it would be valuable for the authors to compare the Tn3 and Transib structures and discuss differences but also possible parallels, with other transposition pathways.

Following the reviewer's suggestion, we first examined the active site configuration of other DDE/D transposase/integrase superfamily members. We found that the region located just downstream of the RNase H-like domain that consists of the switch helix and a loop connecting the last helix of the RNH domain (H_{R3}) with the switch helix has features remarkably conserved in all examined structures (Supplementary Fig. 5). Despite different lengths and sequences, the structural homologs of the "switch" helix are found in all examined transposases in similar orientation relative to the active site of RNH domain such that switch helix is positioned at a similar angle relative to the H_{R3} helix. And in all structures of DNA-bound transposases, the switch helix is in an extended conformation, like the one we find in TnpAS911R-DNA complexes.

In virtually all examined cases, the loop connecting H_{R3} with the switch helix contains two positively charged residues, R899 and R901 in Tn4430 TnpA, which bind cis -DNA (Supplementary Fig. 5). The high conservation of this structural element suggests that it plays a functional role in transposition.

In Transib transposase, rearrangement of the RNase H fold accompanies large rotational movements of the transposase domains that lead to active site assembly and disassembly during the successive steps of the transposition pathway. Despite the lack of primary sequence or structural similarities between the transposases, these large conformational changes are reminiscent of those observed in Tn4430 TnpA structures upon PEC formation. A close comparison of the catalytic domain in the apo form and the DNA-bound activated form shows that in Transib transposase the N-terminal turn of the switch-like a helix unfolds, which re-orientates the catalytic E435 residue, and together with other readjustments of the RNase H fold, promotes the assembly of the active site in trans.

The results of this analysis suggest that catalytic activation upon DNA substrate binding is likely, not unique to Tn4430 as predicted by the reviewer, and that interaction between transposase and cis DNA mediated by the loop connecting H_{R3} with switch helix functions as a conformational signal to activate RNH domain in DDE/D transposases. In Transib, the activation, however, does not involve a large change in folding of switch helix nor refolding of the RNH domain pointing toward the differences in the specific structural pathways of RNH domain activation.

New supplementary figures 5 and 13 were added to demonstrate our conclusions and the text extended to describe them (from line 222): "The RNase H fold insertion domain and the C-terminally located H_S , which together constitute the SFD, are along with the RNH domain the best-conserved regions of Tn3-family TnpAs (**Supplementary Fig. 2**), suggesting a functional role of SFD. Intriguingly, an α -helix equivalent to H_S is present in all currently

available structures of DDE/D transposases irrespective of the transposition mechanism and its position is structurally conserved relative to the RNH domain (**Supplementary Fig. 5**) despite different lengths and sequences. Furthermore, the loop connecting RNH with H_S-like helix carries two positively charged residues, structurally homologous to R899 and R901 in TnpA, that interact with transposon end in cis with exception of Mos1 and MuA transposases (**Supplementary Fig. 5**). These observations suggest that the loop and H_S-like helix are structural and possibly functional conserved elements of the DDE/D transposases similar to RNH domain. “

And from line 269:

“ Another transposase for which structures of apo and DNA-bound forms at different stages of transposition are available is Transib³¹. Unlike in TnpA, activation of Transib is not accompanied by metamorphic refolding of the RNH domain or change in the fold of the switch helix, however, the loop connecting H_{R3} with H_S-like helix, does change the conformation upon activation in strand transfer complex with simultaneous rotation of H_{R3} helix leading to the assembly of the catalytic site (**Supplementary Fig. 13**). This suggests that the association of the loop preceding switch helix with cis-DNA may function as a motif that recognizes binding of cis DNA and activates RNH for DNA cleavage in trans in the other transposes. ”

2. While the cryoEM structures are intriguing and of overall high quality, the manuscript lacks functional validation of the structural findings. A good amount of published biochemical data already support certain structural features, but the functional significance of novel unexpected findings should be tested. For example, is the refolding of the switch helix needed for transposition? Is the surprising “fluidity” of the RNH fold relevant in the natural context (e.g. WT TnpA) and does it contribute to transposition efficiency?

We understand and share the concerns of the reviewer regarding the unknown functional significance of the observed refolding of the RNase H domain and changes in the conformation of the switch helix. While the functional role of these changes in conformation is indeed intriguing, it is not easy to assess from the functional perspective. This is mainly because to assess the functional significance of the changes in the folding we need to be able to interfere with the fold of the “fluid” elements in a rational way through point mutations or larger modifications of the protein. At this moment to the best of our knowledge, the molecular determinants of protein metamorphism remain ill-defined. Therefore, to understand the specific role of changes in the fold, in addition to functional assays, all the mutations will need to be validated structurally in apo and PEC conformations. To accomplish this a separate study will be required that goes far beyond the present report.

This study was focused on the structural investigation of TnpA and observed metamorphism came as a surprising finding. Maybe the title is calling for more functional proof and, in some way, misleading. If it helps, we can change the title to “Structural insight into Tn3 family transposition mechanism”.

Some of the reviewer's questions can be answered based on our structural data:

1) Is refolding of the switch helix needed for transpositions? Yes. In apo conformation, the kinked part of the switch helix prevents the RNH domain from the assembly because it does not have room to fold. Whether it is possible to design a mutant in which switch helix will have extended conformation (like in PEC) already in the apo state and what would be the functional consequence of this we don't know. Answering this question will require a separate study.

2) Is the 'fluidity' of the RNase H domain relevant in a natural context? We think that it is relevant because: (a) Structure of the apo state is obtained with the wild-type TnpA and therefore is expected to be representative of the protein conformation in the cell in the absence of bound DNA. We also know that TnpA expressed in *E.coli* is functional. (b) The complexes of TnpA with transposon ends are obtained with S911R mutant, but their structural signatures suggest that the structures do reflect functional state for the following reasons: the confirmation of switch helix in S911R PEC is similar to wild-type structures of PEC in other transposases; the RNH domain assembles in the right fold compatible with active catalytic domain. Therefore, we believe that the fold 'fluidity' observed in our structures is relevant to the native context.

3) Does 'fluidity' of the RNase H domain contribute to transposition efficiency? It probably mainly contributes to the transposition reliability, and this would be consistent with our proposal that target DNA may bind first. We expect that the binding of target DNA occurs while the rest of the protein remains in an inactive conformation, probably something like the apo-DNA complex (Fig. 6a). In this case it is important that target DNA is not cleaved before transposon ends to bind. That is why refolding of the RNH domain in a conformation in which it cannot be functional at all can be one way to ensure that no premature cleavage of target DNA occurs. And this is very specific to the past-and-copy transposition mechanism. We clarify it better in the discussion section now from line 451:

“Unusual fold of RNH domain in apo state likely serves to ensure that target DNA is not cleaved even with low probability before transposon ends bind. The refolding of the RNH domain into active conformation is thus dependent on the binding of transposon end which refolds switch helix making a room for folding of the RNH domain into the active conformation.”

3. The peculiar rearrangement of the internal folding of RNH domain itself is very surprising and additional functional validation would be helpful. Also, what are the specific features of the Tn4430 TnpA that underlie this structural fluidity? Are these features shared by (some) other DDE transposases?

As mentioned above, there is no readily identifiable signature for protein "fluidity" and/or protein regions amenable to metamorphic folding. However, as discussed in the paper and above (see reply to comment 1), the key to Tn4430 TnpA active site remodeling is its embedding within a frame-like structure that we have called "scaffold" and refolding of the

switch-helix. As described above, the finding that an equivalent to the switch helix is structurally conserved in virtually all members of the DDE/D transposase/integrase superfamily suggests that a similar mechanism may act in other systems. Again, it remains to be determined whether a complete reorganization of the RNase-fold structure as in Tn4430 occurs in the other transposases or conformational rearrangement is limited to a more subtle refolding and switching as observed in Transib. Answering this question requires further investigation, including new structural data on the apo forms or pre-activation states for other DDE/D transposases.

4. The manuscript describes all structures in parallel, comparing selected aspects in four structures simultaneously. This makes the flow rather difficult to follow, and the reader easily gets confused about the different pieces of data. I had to go back and forth in the text and figures constantly to remind myself about the content and appearance of the different structures to grasp and appreciate the various descriptions and discussions. The description of the TnpA IR71st complex is particularly confusing: it is never specifically described really, and it is not clear if the authors consider this finally as an STC or PEC.

In the revised manuscript we improved the overall description of the structures and the flow. Now Figure 2 provides an overview and more complete description of all the structures of TnpAS911R-DNA complexes that of IR71st which we consider very close to STC. The description is given in the section “Architecture of TnpA-transposon end complexes” and the structure is visualized in Figure 2d.

Smaller points:

- The strategy of using a hyperactive variant to capture a stable PEC structure is clever. However, I wonder if the mutation partly biases transposase conformation. An assessment of the S911R apo (or the WT PEC) conformation would be helpful to clarify this possibility.

Above we provided the arguments for why we think that despite the mutation the conformation of PEC or STC-like for the S911R mutant reflects the native conformation: The structural signatures of S911R mutant suggest that the structures do reflect functional state for the following reasons: the conformation of switch helix in S911R PEC is similar to the wild-type structures of PEC in other transposases; the RNH domain is assembled in the right fold which is compatible with active catalytic domain.

We also studied the structures of TnpA^{S911R} in the apo state and the structure of another hyperactive triple mutant W24R/A174V/E740G. For S911R apo state we obtained 2D class averages only but not 3D reconstruction due to strong preferred orientation and for the triple mutant we obtained reconstruction at a resolution of 6 Å (see figure below).

We can see that TnpA^{S911R} exists as a dimer in apo conformation and based on the 2D class averages the shape of the dimer is very similar to the wild-type apo conformation suggesting that TnpA^{S911R} behaves very similar to the wild type. The triple TnpA mutant assembles in a PEC and at 6Å resolution is indistinguishable from the S911R. Given that the sets of mutations activating TnpA are different, we can conclude that the S911R mutation does not bias the protein conformation.

The results shown above are included in another manuscript with a focus on AFM studies of the DNA binding to TnpA titled:

“AFM-based force spectroscopy unravels the stepwise-formation of a DNA transposition complex driving multi-drug resistance dissemination” which will be deposited on bioarxiv within the next 2 weeks and the corresponding correct citation will be included in the manuscript.

We included a brief description of these points in the Discussion section from line 463:

“ Our conclusions regarding the activated form of TnpA are based on the structures of hyperactive mutant S911R. However, structural features of the mutant like the conformation of switch helix conserved in other transposases, the fold of the RNH domain, and the fact that it is catalytically active²⁹ suggest that the mutation does not bias TnpA conformation. Moreover, 2D class averages of the apo TnpA^{S911R} suggest that in apo state its conformation is similar to that of TnpA^{WT} (Fernandez et.al. to be deposited to biorxiv).”

- It would be valuable to analyze and discuss the broader relevance of the results within and beyond the Tn3 family. For example, the authors state that the DNA binding residues are not conserved in the Tn3 family. But are they conserved in Tn4430 copies? And how about the switch helix, is this conserved in the Tn3 family? Are the specific features of the RNH domain that may underlie its plasticity conserved?

We have analyzed the sequences of Tn4430 transposases. Within this subgroup the transposases are very conserved (~ 95% identity). When sequences are compared between subgroups of Tn3 family the conservation is much lower (Supplementary Fig. 2) but the RNH domain and switch helix are among the best conserved regions. The residues interacting with

donor DNA are not conserved between the subgroups. This is consistent with the fact that subgroups recognize IR sequences which are quite different from each other, and it is known that transposases from different subgroups do not complement each other (Refs 10, 28). Therefore, we think that the conservation of IR sequences likely reflects the requirement for a particular structure of DNA double helix in the IR region needed for indirect readout of the DNA sequence. We modified the manuscript accordingly from line 184:

“ Consistent with the high specificity of TnpAs for their respective IR^{10,28}, sequence recognizing residues and corresponding nucleotides, with exception of R44-T36, display modest or no conservation (**Supplementary Figure 2**) and only two pairs R97-G32 and R267-G14 display high covariance between amino acid and nucleotide (**Fig 4a**). Therefore, conservation of the transposon recognition sequence¹⁰ likely reflects the geometric constraints required for matching the DNA backbone to the extended DNA binding surface of TnpA and IR recognition through an indirect read-out mechanism³⁸.”

- *The Figure legends should describe the illustrations more specifically. It is often difficult to guess what exactly is shown.*

The figure legends have been expanded and now describe the figures more specifically.

- *When describing the TnpA-IR71st structure, it should be stated clearly that the target mimicking stem is poorly visible in the EM maps. I was confused about this until the Discussion.*

This change has been introduced in the revised manuscript from line 141:

“

The design of the IR71st substrate does not allow for complete annealing of the 5 nucleotides long single-stranded target DNA segments, and it ,therefore, did not assemble into a canonical Shapiro intermediate³⁶ in which 5 base pairs from the target remain base-paired after staggered strand transfer of the transposon ends. The DNA branch corresponding to the target is mainly disordered starting from scissile cysteine, however, a low-resolution density consistent with DNA fragment is observed adjacent to the RNH domain in the map of TnpA^{S911R}-IR71st (**Fig. 2d**). It was assigned to the target-like branch of the IR71st substrate. “

- *Lines 93-94: Are the domain folds truly novel? Does a DALI search with individual domains give no meaningful hits? Some domains certainly have known folds – please correct.*

To clarify this point we added a Supplementary Table 3 with results of DALI search for individual domains. Also in the revised manuscript the statement about fold similarities have been made more specific line 77:

“Fold similarity search indicated that apart from DBD1, DBD4 and RNH domains, it is constituted of small domains with novel fold (**Supplementary Table 3**)”

- Lines 235-237: I do not understand how the authors arrive to the conclusion that the target DNA plays a role in activation and target immunity. I did not find any data that would support this proposal. Please clarify or remove.

The role of target DNA in activation is inferred from the fact that in presence of only transposon ends bound, the RNH domain is still very dynamic and does not have full expected activity, therefore, we believe that only in presence of target DNA is the RNH domain stabilized in a fully active conformation.

The proposed role of DNA in target immunity is different. It is much more passive. According to our model, once TnpA is bound to target DNA the interaction with an IR end found on the same DNA molecule will be favored over the interaction with IR ends found on the different DNA molecules due to a proximity effect. This would reduce the probability of binding 2 IR ends from another DNA molecule. The interaction of TnpA with 2 IR ends on the same molecule to which it is bound would result in inversion or deletion of the transposon, whereas interaction with one IR end from donor DNA and one IR end from target DNA will not result in transposition. What hyperactive mutants do is that they simply increase a frequency of activation of TnpA upon binding of IR end which also increases the probability of binding 2 IR end from the donor DNA. It is a kinetic effect.

We adjusted the text as follows to improve clarity line 345 Discussion section:

“Binding target DNA to TnpA first may explain the mechanism of target immunity. Once bound, due to the effect of increased local concentration, TnpA would sample sequences on the same DNA molecule with higher frequency as compared to the sequences located on separate DNA molecules or in remote regions of the target DNA. Thus, whenever transposon is already present in the proximal regions of the target, TnpA preferably binds transposon ends on the same DNA molecule. Transposition into the same DNA molecule results in the inversion or deletion of the transposon¹⁶. Hyperactive mutants have a lower activation barrier and bind transposon ends more frequently than wild-type TnpA. Therefore, the net probability of binding to transposon ends on another DNA molecule is higher producing an effect of reduced immunity.”

- Line 258: Why do the authors expect to see a SEC in the TnpAwt-IR100 complex? Please explain.

The SEC was observed in biochemical experiments by EMSA (Nicolas et.al. PNAS 2017) and also by AFM-based force spectroscopy (Fernandez et.al. to be deposited to biorxiv).

- Lines 276-286: I cannot follow the authors model for target immunity. Perhaps I am missing something, but for me it seems that target tethering would lead to “local hopping” rather than target immunity.

Target tethering will lead to local inversion or deletion rather than integration of the transposon but at the same time, this will reduce the probability of interaction with donor DNA and, therefore, that of productive transposition as described above.

Reviewer #2 (Remarks to the Author):

This is a very interesting paper that describes for the first time a series of DNA transposase and transposase/DNA complex structures from the Tn3 family. One aspect that makes the result novel is that this is the first time that a full-length replicative DNA transposase has been captured both in the apo and transposon DNA end-bound forms. A second novel aspect is that the transposase is already a dimer in the apo form and remains as a dimer in the transposon end-bound forms (PEC) as well, without forming higher order assemblies as observed for other similar systems (eg. phage Mu transposase). Furthermore, as the authors point out, the architectures of the assemblies and the two independent dimerization interface suggest that it is very likely that the target DNA binds first and the transposon ends are subsequently captured by the transposase/target DNA complex. While this is only a suggestion given the lack of clear target DNA density, it is very plausible and consistent with the available data, providing substantial mechanistic insight into Tn3 transposition. Given the importance of mobile genetic elements, and Tn3 in particular, in the spread of antibiotic resistance, I believe the results should be of wide interest.

That said, the paper needs several major issues to be addressed before publication.

1. Throughout the text when describing the results, it is not clear which of the four solved structures authors are referring to. As there are three different PEC structures, a consistent and definitive notation should be applied and used whenever a particular structure is analyzed or discussed. When a particular feature is common between all three PEC structures, this could be noted as "in all three PEC structures".

In the revised manuscript we have improved notation and description of the TnpA-DNA complexes and refer to TnpA^{S911R}-IR100 and TnpA^{S911R}-IR48 as PEC structures and to TnpA^{S911R}-IR71st STC like structure. The overall description of the conformations and flow has also been improved.

2. The paper needs much better figures. The assemblies are complexes with many domains and protein/protein and protein/DNA interactions. The chosen representations just do not do the job, are quite dark, and in particular Fig1d, Fig5a,b are not sufficient and Fig4a not at all useful. I very strongly urge the authors to include a schematic representation such as Figure 2a in Montano et al. (2012) Nature 491, 413 to show the essence of the assemblies. I should also like to point out that having a 3D cartoon representation in the same orientation adjacent to the schematics (as in Montano et al.) gives an unambiguous perspective and provides clear understanding. The use of a reasonable amount of fog on the structure figures (e.g., Fig 2b,c Montano et al.) is very useful for depth cueing.

We have re-made all the main text figures. Figure 1 was split into 2 figures to better visualize all 3 structures of TnpA-DNA complexes and Fig4a was removed and a reference to Supplementary Video showing conformational changes is made instead.

We hope that a combination of better figures with improved annotated videos can help to better visualize the organization of the domains in TnpA. We felt that our attempts to make cartoon representation were not successful enough to explain the 3D relation between the domain positions therefore they were not included in the updated figures.

3. The authors write about and display the conformational morphs between apo and PEC states. However, I find a logical inconsistency as the model in Fig5c describes an apo -> target bound -> target+transposon end bound progression. Therefore, I do not understand the point of discussing much about the apo -> PEC conformational transition as this does not appear to be really relevant, unless of course I'm missing something.

It is correct that in the proposed model the TnpA conformation that we observe with the bound transposon ends does not constitute part of the transpososome assembly pathway. However, there are 2 reasons why we think that displaying the conformational morphs is still useful. 1) It is a good way to visualize the conformational differences between 2 structures. 2) The structural properties of the TnpA-IR complexes suggest that their conformation is compatible with target DNA being bound to the complex which suggests that binding of target DNA would likely not cause the major conformational changes, at least this is our expectation.

To avoid misinterpretation of the morphs we clearly state in the legend of the Supplementary Video 3 that “The morphs are shown only to visualize the differences in the conformations between two structures and should not be considered as visualization of the real conformational transition taking place during the transpososome assembly which we expect will include an intermediate step of target DNA binding”.

4. I do not understand why the DD dimerization domain was not docked into the apo density. This dimerization interface is highly hydrophobic, and I would think that this is primarily responsible for holding the apo dimer together. "Protein-protein dimerization interfaces are very similar between apo and PEC conformations" yet the apo structure has no coordinates for the DD. I believe authors should consider that the apo form opens up at CT, while DD holds together to capture the target DNA.

We have now fitted the DD domain in the apo map and updated the apo model. The mobility of the DD domain in apo state is quite high not allowing for a reliable *ab initio* modeling therefore initially we did not build DD in the apo density. Materials and Method section, Figure 1 and Supplementary Videos 1 and 3 were updated accordingly.

5. Please compute the sizes of both CT and DD dimerization interfaces.

The sizes of interfaces have been added as Supplementary Table 4 and are also now mentioned throughout the text.

6. If target DNA is indeed bound before the transposon ends are bound, authors should show how and why target DNA would not sterically conflict with the DNA flanking the transposon. This is because, when transposon DNA is bound, it has a flank and it has to go somewhere, and in fact two of the PEC structures indicate where that would be. This is a significant mechanistic point, as there are other known DNA transpososome structures where target DNA clearly binds in the transposon DNA flank binding site, unambiguously establishing the time sequence of the events. If Tn3 is different, transposon flank and target DNA must coexist even if transiently.

This is a very valuable mechanistic comment. Indeed, the assembly of transpososome followed by formation of Shapiro intermediate requires 2 regions of donor DNA and target DNA to be assembled. To demonstrate that this is sterically possible in TnpA we built a putative target DNA model into TnpA^{S911R}-IR71 complex. To do so we placed helical DNA into the low-resolution density observed in TnpA^{S911R}-IR71 complex and assigned to DNA branch mimicking target DNA (Fig. 2d) and placed double stranded DNA in the opening between TnpA protomers and the DD by aligning phosphates close to scissile OH groups of the transposon ends. Then we connected the pieces creating a bent target DNA model followed by local minimization in Coot to avoid clashes. Even though this model is very putative it allows us to make conclusions regarding possibility of binding target DNA to the PEC conformation of TnpA. 1) We see that double stranded DNA can be placed into the opening without major clashes (conformation of some side chains had to be changed). 2) Number of nucleotides between putative strand transfer sites is equal to 5 as expected. 3) The outer flanking segments of donor DNA do not clash with target DNA.

We added Supplementary Figure 14 showing the target DNA model. And the following text was added to Discussion section on line 284:

“To further support these conclusions, a putative model of target DNA was added to TnpA^{S911R}-IR71st complex (**Supplementary Fig. 14**) such that double-stranded DNA threads through the opening and fit the density assigned to the target-like DNA branch (**Fig. 2d**). The model of target DNA does not clash with TnpA or outer flanking segments, and it is strongly bent which is commonly observed in transposases^{31,33,34}.”

7. I believe authors should state clearly that the structural results presented and the resulting model is inconsistent with the model for the transpososome assembly pathway proposed in Nicolas et al. 2017. The inconsistency goes beyond the notion of the monomeric nature of apo form of the transposase.

We added the following sentence in the discussion section Line 267:

“The structures of TnpA in the apo state and paired with transposon ends are dimeric. This observation falsifies a previously proposed model in which active transpososome assembles from TnpA monomers²⁹.”

Minor comments:

Line 80. "The scissile phosphate" needs a figure. While I understand that part of the active site is disordered in all of the PEC structures, a reader would probably like to see the relative position of the scissile phosphate to the two visible carboxylates.

A panel e was added to Figure 2 showing the scissile phosphate and the resolved residues of the active site.

Line 81. Please do not use "cross-reactivity". The common terminology in the field is trans activity, in which the protomer's active site processes the other end relative to where its site-specific DNA binding domain is bound.

We would like to thank the reviewers 2 and 3 for pointing this inaccuracy, we replaced the term line 100:

“This cis-trans arrangement, wherein one subunit recognizes and binds to one transposon end in cis (cis-interaction and cis-DNA), while catalysing DNA cleavage and strand transfer in trans on the partner end (trans-interaction and trans-DNA) is a convergent feature of most characterized DDE/D transposases despite their structural heterogeneity”

Line 92. "Structurally unique insertion domain". I do not see the results of a Dali search run with the coordinates to establish uniqueness. This has to be done. Are the authors certain that there is no similarity with the Rag1, Hermes and P element insertion domains?

We added Supplementary Figure 5 which compares the insertion domain structures of TnpA with other known transposases and shows that structure of TnpA insertion domain is unique. The result of DALI searches for the individual domains are now provided as Supplementary Table 4. Despite similarity found by DALI for all the domains, in most cases the folds are matched by completely different proteins which have similar arrangement of some alpha-helices but don't have similar length of the helices and have different loop arrangement when overlaid structures are visually inspected. This is reflected by the fact that for those domains the best search matches correspond to functionally completely unrelated proteins.

The corresponding sentence in the main text is now modified to more accurately reflect this fact. Line 77 :

“Fold similarity search indicated that apart from DBD1, DBD4 and RNH domains, it is constituted of small domains with novel fold (**Supplementary Table 3**).“

We once again inspected the architecture of complete RAG, Hermes and P element transposases and their respective insertion domains into RNase H domain and we don't find structural similarity between them.

Line 94. Did a Dali search provide no homologs to DBD2 or DBD3?

We could not find any in the list of the closest matches found with DALI (Supplementary Table 4). The helix arrangement in these domains might be perturbed enough to fail the structural alignment. The similarity between TnpA DBD4 and DNA binding domain of Tn5 for which DALI does not find the correspondence and which are also very difficult to align by LSQ or secondary-structure-based alignments were discovered by visual inspection and initially manual alignments followed by automated alignment with carefully selected structural regions.

Line 99. As in major comment 4, please dock the DD domain into the low resolution density or give a clear reason why this cannot be done given that its structure is known from the PEC structures.

The DD has been docked and model updates as requested.

Line 104. "unusual dimerization interface". Unusual in what sense?

Line 105. "important consequences for the transposition mechanism". What consequences in particular?

We removed this sentence and return to both points in discussion where it is clearly explained why the interface is unusual (because it has 2 dimerization areas).

Line 108. "cis-DNA". This is insufficient. Please define this DNA better for the general reader not familiar with the terminology of the field.

The notion of cis- and trans- interactions is now defined on lines 100-104: "This cis-trans arrangement, wherein one subunit recognizes and binds to one transposon end in cis (cis-interaction and cis-DNA), while catalysing DNA cleavage and strand transfer in trans on the partner end (trans-interaction and trans-DNA) is a convergent feature of most characterized DDE/D transposases despite their structural heterogeneity."

Line 116. "reduction in the bending angle". Reduction relative to what?

The sentence was improved. Now it reads as follows: Line 123 "This suggestion is supported by the reduction in the substrate bending angle from 72 ° for IR100 to 54 ° for IR48, in which the interaction of outer flanking segment (5 bps) with DBD3 is reduced".

Line 122. "broken transferred strand". Please explain why the transferred strand is broken. I think this is because of how the IR71 substrate was assembled, but this might not be clear for

the general reader. I also note that in Methods the complex assembly section (after line 335) is only cursory and not detailed enough as written for a general reader. The relative protein DNA concentration ratios are missing for the PEC complexes.

Indeed, by the broken strand we meant the fact that in the construct the outer flanking segment of the branched IR71st substrate is separated from the IR sequence. It has been now clarified in the manuscript as follows (line 129):

“In the structure of the post strand transfer-like complex, TnpA^{S911R}-IR71st, in which the transferred strand of the IR end is disconnected from the donor and joined to the target DNA (**Supplementary Fig 3a**)”

The materials and method section describing preparation of complexes is more detailed now (line 453):

“The protein and DNA concentrations during the complexation of TnpA^{WT} with IR100 were 0.5 and 2.5 mM while the complexation of TnpA^{S911R} with DNA was performed at concentrations of 2.5 and 10-25 mM, respectively. Due to low stability, TnpA^{WT} was complexed with IR100 for 1 h at 4° C after which it was plunge-frozen.”

Line 125. "making transposition irreversible". Please expand to explain this to the uninitiated.

This section of the manuscript was significantly modified. We explained the irreversibility as following now lines 132:

“This repositioning of the outer flanking segment suggests that stress release upon scissile bond cleavage is necessary to avoid clashes between the transposon ends and the target DNA, enabling it to approach the attacking 3’-OH group of the transposon end for the strand transfer reaction. The bending of outer flanking segments is also needed to free the room for and avoid clashes with target DNA that should simultaneously bind to the active site (see below). The role of DNA bending at the ends of the transposon is thus mechanistically distinct from the role generally evoked for target bending which is to make the theoretically reversible strand transfer reaction energetically favorable in the absence of external chemical driving force^{31,33,34}.”

Line 126. "DBD3-OFS interaction" is labeled as DBD4 in the legend of Fig 2d.

The panel names are given after the name of the corresponding domain which is why it may appear confusing.

Line 127. "The interaction is also..." Please rewrite this sentence, explain it better, and fix the punctuation.

This part of the manuscript has been rewritten the sentence reads as follows (line 159) :

These interactions are promiscuous permitting DNA binding to DBD3 in different positions and orientations (**Figs. 2f,g, Supplementary Video 2**).

Line 134. "DNA-binding domain surface with BEN domains". How was this similarity found? Was a Dali search run? What were the results?

We added Supplementary Table 4 with results of DALI similarity search. DBD1 was among 2 domains that displayed actual structural match as one of the top search result.

Line 176. Given that it is the nature of the mutation (S911R) that is informative - not simply that it was mutated - it would be nice to see Extended Data Fig 8 modified to specify the mutations, not just indicate their positions.

The Supplementary Fig. 11 has been modified as suggested by the reviewer.

Line 198. "D751" I see no evidence of D571 in any of the PEC coordinate files, so I'm not clear how statements about it can be made. Please clarify.

The respective paragraph was modified. The density for D571 is absent but present for D750, suggesting that D751 is found close to its position predicted by AlphFold2 model. The modified paragraph reads as follows (line 250):

“Among them, E881 that sits on H_{R3} has a well resolved density and it faces scissile bond of C1 (**Fig. 2e**) on the transferred strand (**Figs. 2f, 4d**), whereas low-resolution density is present for D679 while the density for D751 is absent (the density stops at residue T750) in all three structures of TnpA-DNA complexes.”

Line 200. "additional factors are required". This is mysterious, so please explain what additional factors could be involved.

What we meant here is that although TnpA is the only transposon-specific protein responsible for mediating transposition and self-immunity, we can not rule out the possibility that a yet unidentified host factor is involved in the process by facilitating access to the target.

We modified the manuscript to make it more clear. From line 332:

“ The binding might be enabled by transient disruption of dimerization interface mediated by CT fragments and occur spontaneously or upon interaction with target DNA, which might need to be in a specific conformation in order to become a permissive target.”

Line 202. "unique regulatory mechanism". Please explain how this is a regulatory mechanism.

We explain this in the revised manuscript in Discussion section from line 370:

« Mu transposition is mediated by two main proteins: the transposase MuA and the ATP-dependent target binding protein MuB also involved in transposase activation and target immunity. The Mu transpososome is an oligomeric complex in which the core transposase is constituted by a tetramer of MuA³³. MuA also stimulates ATP hydrolysis by MuB promoting its dissociation from adjacent target DNA regions making them immune for transposition. In Tn3-family, TnpA is the only transposon-specific protein involved in both transposition and target immunity. »

Line 215. "MuB". Please mention a main feature of MuB is that it is an ATPase and the key factor in mediating target immunity in Mu replicative transposition. The point is that Tn3-like elements appear to display target immunity without an ATPase.

Now it has been mentioned as specified in the response to the previous comment.

Line 220. "unique and totally differ from those of Mu". This is imprecise. Please explain how it differs.

This has been answered now together with the response to the previous comment.

Line 273. "cis switch helix" This is not defined, so it is not intelligible. Although it sounds interesting, it needs a better explanation and perhaps a figure somewhere?

This was confusing indeed. We have modified this paragraph and now the description of the mechanism should be more clear (line 335):

“The binding of target DNA is followed by the sequential binding of the transposon ends associated with the conformational changes in TnpA protomers. They cause the switch helix of the corresponding cis protomer to refold inducing folding of the RNH domain of the same protomer.”

Line 279. "transposon is already present in the target". Is the presence of the entire transposon required for immunity?

The presence of a single end of Tn4430 is sufficient for the immunity although 2 ends in opposite orientations are more efficient as was shown for by *Lambin et.al. 2012*.

Line 280. "transposon recognition motif". Please define this with the terminology already used in the paper.

The “*transposon recognition motif*” has been replaced with “IR end”

Line 284. "Whichever the..." Please rewrite this sentence. I have no idea how transposition can be effective without all substrates present.

We removed this sentence in revised discussion section.

Line 400. "were predicted using AlphaFold2". I do not understand this. I assume that AlphaFold2 predicted the structure of the entire RNaseH-like catalytic domain, so why wasn't such a model placed in the PEC structures based on alignment between strands/helices that are visible in the density? Or was this tried but resulted in clashes? If so, what would be the explanation? I would like to know what sequence was input to AlphaFold2, was it the entire transposase, or only the RNaseH-like catalytic domain? What were the results, how does that compare with the experimental structure?

We predicted the entire structure of TnpA using AlphaFold2 and overlay of predicted model and the model build into the density is shown in Extended Data Fig. 9c.

Now we clarified this in Materials and Methods section Line 524: To build a poorly resolved b-strand of the RNase H domain in TnpA^{S911R}-DNA complexes were predicted the entire structure of TnpA using AlphaFold2⁴¹ (**Supplementary Fig. 12c**) and fitted complete predicted RNase H and scaffold domain into the density as a rigid body. Next, regions for which density was absent were removed from the model. ”

Table 1. Ramachandran plot. My hope is that not almost 100% of all residues are in the disallowed regions. Please fix.

Apologies, corrected.

The movies. While I like the movies, they should have legends describing what are we looking at. Also, can they please be retraced at higher resolutions? Please use the "viewport" command in pymol, with some suitable large numbers while keeping the correct aspect ratio.

We retraced and annotated the movies.

Reviewer #3 (Remarks to the Author):

This manuscript describes the long-awaited and quite surprising structure of Tn3 transposase in complex with DNA substrates. Beyond its historical and medical significance, Tn3 transposase has a feature that could make it more useful than other transposases for, say, transposon library construction: transposition and target immunity functions within a single protein. Overall, I am very enthusiastic about this major advance. The models appear to be carefully built and refined, and the results are thought-provoking – particularly the idea that in this system, the target DNA may bind BEFORE the transposon ends, and that that might be a key feature in target immunity. However, I found the presentation rather confusing.

Substantive comments:

1) Table 1 has typos that slander the nice models: it says 98% of the residues are in the

dissallowed Ramachandran plot regions, rather than in the favored regions (as shown in the validation report and the fact the model has nice secondary structure).

Thank you for drawing our attention to this annoying typo. It has been corrected.

2) The beginning of the Results section points out that substrate IR71st included target branches, but the lack of the target DNA in the figures isn't explained until close to the end. I found this very confusing.

In the revised manuscript the flow and presentation of structures is improved. Now the densities and models for all 3 complexes with DNA are shown in Figure 2. The description of the IR71st structure is also complete in the result section now line 141:

“The design of the IR71st substrate does not allow for complete annealing of the 5 nucleotides long single-stranded target DNA segments, and it ,therefore, did not assemble into a canonical Shapiro intermediate³⁶ in which 5 bps from the target remain base-paired after staggered strand transfer of the transposon ends. The DNA branch corresponding to the target is mainly disordered starting from scissile cysteine, however, a low-resolution density consistent with DNA fragment is observed adjacent to the RNH domain in the map of TnpA^{S911R}-IR71st (**Fig. 2d**). It was assigned to the target-like branch of the IR71st substrate.”

3) Is the target in the IR71st complexes still covalently attached? Two reasons for wondering if it is not are: (1) Mu transposase, if given a similar substrate, has trouble figuring out which branch was supposed to be target vs. flanking donor DNA, and removes some of the target branches as hairpins (see Mizuuchi ... Mizuuchi PNAS 2007) and (2) the IR71st model has nice low B factors up to and including the 3' end of the transferred strand, then the DNA just disappears – exactly what one would expect if the target had been accidentally cleaved off as in that 2007 paper.

The complex between IR71st and TnpA was prepared in the absence of divalent ions Mg²⁺ or Mn²⁺. Under these conditions, TnpA does not cleave the IR71st substrate, therefore we expect the target branch to be still present in the complex. The density does completely disappear after the 3' end of the transferred strand. We interpret it as the high mobility of the strand which is not constrained by other interactions apart from weak interaction with the RNH domain. We did not observe the additional density next to the RNH domain with any other constructs which suggest that that density does come from the target-like branch of DNA.

In the revised manuscript the description of the density is made more clear (line 141): “The DNA branch corresponding to the target is mainly disordered starting from scissile cysteine, however a low-resolution density consistent with DNA fragment adjacent to RNH domain is observed in the map of TnpA^{S911R}-IR71st (**Fig. 2d**). It was assigned to the target-like branch of the IR71st substrate”.

4) p6 – idea that bending of the flanking DNA could contribute to irreversibility: while it may

be tempting to draw the analogy to the role target DNA bending in irreversibility, I don't think that makes sense. The initial cleavage step is a very favorable hydrolysis reaction that is irreversible anyway (you'd need ATP and ligase to religate that nick). The strand transfer step is where irreversibility requires an explanation, because that is not "downhill" in terms of high-energy chemical bonds. I would guess that the reason for bending the flanking donor DNA is to prevent steric clashes with the target DNA.

We agree. Our original proposal was wrong indeed. We appreciate your suggestion which is also consistent with the model of target DNA that we built to demonstrate the possibility of passing double stranded DNA through the opening in between the protomer bodies and the DD (Supplementary Fig. 14).

Upon donor DNA cleavage, the outer flanking segment rotates away from the scissile bond which may allow target DNA to move closer to the attacking 3'-OH group.

The respective part of the manuscript has been modified a lot and now reads as follows (from line 132):

"This repositioning of the outer flanking segment suggests that stress release upon scissile bond cleavage is necessary to avoid clashes between the transposon ends and the target DNA, enabling it to approach the attacking 3'-OH group of the transposon end for the strand transfer reaction. The bending of outer flanking segments is also needed to free the room for and avoid clashes with target DNA that should simultaneously bind to the active site (see below). The role of DNA bending at the ends of the transposon is thus mechanistically distinct from the role generally evoked for target bending which is to make the theoretically reversible strand transfer reaction energetically favorable in the absence of external chemical driving force."

5) Same argument against the logic of p6 lines 129-130: initial hydrolysis is expected to be irreversible no matter what. The key problem for this type of transposase at this step is probably to get the flanking donor DNA out of the way of the target, which has to access the same active site.

We removed this sentence.

6) I think the authors are really "onto something" regarding the connection between Tn3 transposases's mysterious target immunity and the idea that it binds target 1st rather than last. However, the discussion of possible mechanisms on p. 13 is confusing. By "tethering" do the authors suggest that the transposase initially forms a sliding clamp on non-specific (target) DNA? What is the "transposon recognition motif" referred to? (the transposon ends or something else?) What exactly is meant by avidity? Cartoons may help. Could it be that sliding of the transposase complex along target DNA results in a non-productive complex (and perhaps dissociation) if it slides onto a transposon end, thus engaging the 1st couple of DBDs while still binding part of the same DNA molecule in the target-binding spot, but

results in productive transposition if transposon ends that are on a separate DNA segment join the complex in trans?

We improved language in the model description part. Here we use word binding instead of tethering. Sliding clamp is a possibility but transposase does not need to slide for the mechanism to work. We replaced ‘transposon recognition motif’ with transposon end. Avidity was also not an accurate word. It is replaced by “effect of increased local concentration”.

Updated text reads as follows line 345:

“Binding target DNA to TnpA first may explain the mechanism of target immunity. Once bound, due to the effect of increased local concentration, TnpA would sample sequences on the same DNA molecule with higher frequency as compared to the sequences located on separate DNA molecules or in remote regions of the target DNA. Thus, whenever transposon is already present in the proximal regions of the target, TnpA preferably binds transposon ends on the same DNA molecule. Transposition into the same DNA molecule results in the inversion or deletion of the transposon¹⁶. Hyperactive mutants have a lower activation barrier and bind transposon ends more frequently than wild-type TnpA. Therefore, the net probability of binding to transposon ends on another DNA molecule is higher producing an effect of reduced immunity. ”

7) p8, bottom and extended data Fig. 8: do hyperactivity and target immunity generally correlate? And are some of these mutations at protein-protein interfaces in the full complex?

So far, yes. The mutants we were initially selected for their immunity impairment phenotype (Lambin et al 2012). Later it as shown in vitro assays (Nicolas et al 2017) that they have higher activity than TnpA^{WT}.

We mention hyperactivity in the discussion of the immunity mechanism line 351:

“Hyperactive mutants have a lower activation barrier and bind transposon ends more frequently than wild-type TnpA. Therefore, the net probability of binding to transposon ends on another DNA molecule is higher producing an effect of reduced immunity”

9) Mu transposase also has a protein-protein interaction segment blocking “escape” of the target DNA, but Mu is known to bind target last. This might be worth discussing, especially because the dimerization domain seems to be disordered in the apo structure.

The C-terminal end of MuA contains a specific helical domain termed IIIa (alpha) that has multiple and different functions depending on the subunit forming part of the active tetramer. In the outer-most, R1-bound subunits I, IIalpha plays a role in target binding and traps it within the target capture complex by forming a coiled-coil dimeric interface between the left and right R1 protomers. This function of target trapping is thus equivalent to the role proposed here for the TnpA CT domain. The main difference with TnpA is that in TnpA DD

is statically dimeric. It has very hydrophobic and large surface (1600 Å²) which prevents it from the dissociation. Its higher mobility in apo state is likely caused by changes in the structure of a long α -helix that connects DBD2 to DD. In apo state part of it becomes flexible leading to poor density. The density for DD is still present even though at lower level. Therefore, it is very unlikely that target binds from the side of DD like it happens in Mu transposase.

10) p. 6, top refers to a “box 1” in Fig 3 – is that “box A” in the figure?

Yes, it should have been “box A”. The typo has been corrected

Suggestions (these are not demands) for making the manuscript easier to understand:

1) if possible, redo many of the figures with better depth cueing (fading out in the back) and without shadows – for a structure this big, the shadows just add visual noise.

All the main text figures have been redone.

2) In the 1st paragraph of results, mention that the target DNA is only partially visible in the IR71st complex, and that there is a hint of target visible in some of the “apo” complexes – then just say it will be discussed later. That will keep the reader from wondering what happened to it.

We changed the flow in the first 2 sections of the paper and better explained the density and the structure of IR71st complex. From line 144:

“The DNA branch corresponding to the target is mainly disordered starting from scissile cysteine, however, a low-resolution density consistent with DNA fragment is observed adjacent to the RNH domain in the map of TnpA^{S911R}-IR71st (**Fig. 2d**). It was assigned to the target-like branch of the IR71st substrate.”

3) In figures such as 1d, explain what the white blob is (electron density or space-filling outline?)

The figure has been changed along with figure legend. We hope that it is more clear now.

4) Cut down on the number of acronyms the reader has to memorize to follow the paper. Unless this journal is too strict about character limits, spell out things like “paired end complex,” “single end complex” and “outer flanking segment.” (the protein already has 10 domains, each with its own acronym).

The number of acronyms has been reduced as per suggestion of the reviewer.

5) The color key in Figure 1 says bright yellow is donor DNA – but that seems to be referred to as OFS in other places

In the new Figure 2 this has been corrected.

6) I'm sorry but I've started at Figure 4a for a long time, and I still can't follow what is going where except for DBD1 (which should be labeled as such). Perhaps showing the models side-by-side rather than superimposed would help.

We removed this panel in the Figure5 (new numbering). In the text we refer to the Supplementary Video3 that demonstrates more clearly the conformational differences between the 2 states.

7) cross-reactivity is confusing because that term can also mean, say, enzyme 1 also works on enzyme 2's substrate. Isn't this usually referred to as "catalysis in trans"?

We replaced the term line 100:

“This cis-trans arrangement, wherein one subunit recognizes and binds to one transposon end in cis (cis-interaction and cis-DNA), while catalysing DNA cleavage and strand transfer in trans on the partner end (trans-interaction and trans-DNA) is a convergent feature of most characterized DDE/D transposases despite their structural heterogeneity”

8) a figure comparing the RNaseH domains from these structures to some seen in previous DDE enzymes would be helpful.

We added Supplementary Figure 5 that compares DDE domain and switch helix between TnpA and other transposases with known structure.

9) Work with a native speaker of English to polish the writing.

We improved English and will have the manuscript checked by a professional editor for the final version.

REVIEWERS' COMMENTS

Reviewer #2 (Remarks to the Author):

In this revised version, the authors have done a good job to address most of the concerns of the referees. However, the manuscript still has a number of inaccuracies and some annoying mistakes that have to be fixed.

Specifics:

Lines 74, 75. SFD? Do you mean SFC?

Line 76: Supplementary, not extended, I believe.

Line 83: I think rather than Supplementary Fig 4c, Fig 3 is meant here.

Lines 88, 89. I think it is a good idea to designate the complexes with the DNA substrates used. The trouble is that throughout the paper the "paired end complex" designation is used and it is not clear which complex this really refers to. Why not use the designation as promised in line 88 "further referred"? So while authors attempted to address reviewer 2's comments, the result is not fully successful.

Line 90: "scissile cytosine" There is no such thing. How about: "The cytosine with the scissile phosphate".

Line 90. "prefigures" This word generally means something else. How about: "It approximates (or mimics) the strand-transfer product".

Lines 95, 96. How about inverting the order of these two sentences, as it would make more sense to talk about binding first and make the statement about the complex after.

Lines 100-102. In Fig 2e, the color code for pink is defined "as scissile bond" yet all covalent bonds of the cytosine are colored like that. How about introducing a color for the actual scissile phosphate, or perhaps just have an arrow pointing to it?

Lines 116: "paired end complex"? Which complex is this?

Line 120: "In paired end complex, TnpAS11R-IR100". Is the IR100 complex the paired end complex or is it one of the paired end complexes? Confusing.

Lines 139-141: Here the authors attempt to address reviewer 3's comment, but again the result is not fully successful. "Theoretically reversible strand transfer reaction energetically favorable". First, the strand transfer reaction is not only theoretically reversible, it is actually reversible. As the reviewer pointed out, this reaction is neither favorable nor unfavorable as the substrate and the product are at about the same energy levels, so it can go back and forth on its own. This is different from the hydrolysis reaction of breaking the DNA chain that will not go backward on its own. How about: "...evoked for target bending which is to prevent the strand-transfer reaction reversing by moving the product 3'OH group sufficiently far away from the scissile phosphate".

Line 143: "Didn't assemble into a canonical Shapiro intermediate". I have no idea what point the authors are trying to make. In the Shapiro intermediate, the broken target strand does not have the ends of the target site as correctly shown on Fig 1a.

Line 148: In Fig 2d, it is not really visible to which domain the disordered target density is adjacent to.

Line 160: How about "The lack of specificity of these interactions are permitting..." rather than "promiscuity"

Lines 205, 208: I believe the authors meant to refer to Fig 5b not 4c.

Lines 211, 212: Can authors state that all hyperactive mutants impair immunity?

Line 220: "The RNaseH fold insertion domain" I suggest a change: "The insertion domain that interrupts the RNaseH fold"

Line 262: It is not clear what distinguishes Tn5 among all other known transposase structures and what core is referred to. What numerical measures justify this choice?

Lines 264-266: "This indicates..." I would suggest leaving this sentence out. "remarkably different evolution" is not something I can parse.

Lines 299-300: "...thread through the opening as a double-stranded break". I have no idea how a break would thread through and even if it could what would make such a break?

Lines 333-341: I think this section is not very clear. The authors are probably onto something, but it needs much clearer writing. What I can imagine is a scenario in which the target DNA, if it contained a Tn3 end, could be bound by the site-specific DBD domains. As they are now engaged with the target DNA, they will not be able to bind the real transposon end to be transposed. Perhaps the ability of the transposase to reconfigure itself to the extent as it seen in this work has also something to do with the reconfiguration needed to mediate immunity as well.

Lines 367-369: "highly concerted and regulated because...can damage...survival of the cell.." I think this is not the case. Tn3 transposition, given that it is targeted non-specifically, can damage anything it touches including the chromosome and can lead to cell death no matter how highly regulated it is. From the point of view of the transposon, there are certainly advantages of being regulated and concerted, but I would not mix in cell survival.

Reviewer #3 (Remarks to the Author):

The authors have nicely strengthened and clarified the paper. My only significant remaining concern is the explanation for target immunity. After extensive pondering, here are my thoughts:

1) It is OK to just say that it is not fully understood. The system has many variables and to the best of my knowledge, nobody has managed to get full transposition to work with purified components, so we don't even know what all the variables are.

2) There is an apparent internal contradiction in the explanation between these two statements: "whenever transposon is already present in the proximal regions of the target, TnpA preferably binds transposon ends on the same DNA molecule" and "Therefore, the net probability of binding to transposon ends on another DNA molecule is higher producing an effect of reduced immunity."

3) The author's hypotheses would explain a preference for non-productive intramolecular (that is, intra-plasmid) transposition over intermolecular transposition, but that seems like an unlikely biological strategy for such a successful transposon. They don't explain why a target without transposon ends would be favored over one with transposon ends.

4) As a side point, the phrase "Transposition into the same DNA molecule results in the inversion or deletion of the transposon" needs re-wording to explain that it is the plasmid backbone that suffers

inversions or deletions, not the transposon itself.

5) My best guess is that this system is conceptually similar to the Mu paradigm, but in this case those concepts are implemented very differently. Perhaps: (a) target-bound transpososomes are somehow dissociated when they encounter a single end; (b) this is in competition with (but faster than) the binding of two transposon ends (that end-binding is slow is supported by the not-fully-docked WT + transposon ends structure referred to, and the need for activating mutations); (c) this gives time to build up a differential distribution of target-bound transpososomes on targets that do not contain transposon ends. The overall concept is similar to the idea for Mu that before transpososomes are fully formed, end-bound MuA triggers dissociation of local MuB, such that by the time a proper Mu transpososome has formed, only distal targets are bound by MuB. In the Tn3-family case, exactly why target-bound transpososomes would tend to dissociate when they encounter a single transposon end would remain to be explained. Also, without an energy source my hypothesis would seem to be a Brownian ratchet that violates the laws of thermodynamics

(https://en.wikipedia.org/wiki/Brownian_ratchet), but perhaps ATP-burning chaperones are involved.

6) It seems easier to explain target immunity if the target binds 1st, and that agrees with the architecture of the transpososomes seen, but it does raise the question of why the authors don't see "target"-bound particles in which duplexes intended as transposon ends are instead being seen as target DNA.

Miscellaneous small things:

1) Figure 1 b abbreviates scaffold as SCF but the text uses SFD.

2) P146 – replace cysteine with cytosine

3) Lines 276 – 278: does it really falsify the previous work, or do the different results reflect different protein concentrations used in the different experiments? (that is, could mass action be driving dimerization in the structural work?)

We thank Reviewers for having carefully re-examined the revised version of our manuscript and for having given us helpful suggestions to still improve it. We found them pertinent and have taken all of them into account. Below we provide responses to the comments and the list of modifications we made during the revision.

Reviewer #2 (Remarks to the Author):

In this revised version, the authors have done a good job to address most of the concerns of the referees. However, the manuscript still has a number of inaccuracies and some annoying mistakes that have to be fixed.

Specifics:

Lines 74, 75. SFD? Do you mean SFC?

The inconsistency has been rectified and SFD replaced with SCF throughout the manuscript.

Line 76: Supplementary, not extended, I believe.

The typo has been corrected.

Line 83: I think rather than Supplementary Fig 4c, Fig 3 is meant here.

We double checked the text and our reference to Supplementary Fig. 4c on line 83 is correct.

Lines 88, 89. I think it is a good idea to designate the complexes with the DNA substrates used. The trouble is that throughout the paper the "paired end complex" designation is used and it is not clear which complex this really refers to. Why not use the designation as promised in line 88 "further referred"? So while authors attempted to address reviewer 2's comments, the result is not fully successful.

To make it clearer we now defined which structures correspond paired end complex (PEC) and which to strand transfer-like complex on lines 88-96. When we mention all TnpA-DNA complexes, we refer to them as "all IR-bound complexes". We also use abbreviations of PEC for paired-end complex, which is commonly accepted in the field. To reduce the number of abbreviations, we stopped using abbreviation CT for C-terminal domain in the text, apart from figures where this abbreviation is explained in the corresponding figure legend.

Line 90: "scissile cytosine" There is no such thing. How about: "The cytosine with the scissile phosphate".

To avoid ambiguity between donor and target scissile phosphates, we replaced "scissile cytosine" with "the IR 3'-end cytosine" which is more accurate term than "cytosine with the scissile phosphate".

Line 90. "prefigures" This word generally means something else. How about: "It approximates (or mimics) the strand-transfer product".

"Prefigures" is replaced with "mimics".

Lines 95, 96. How about inverting the order of these two sentences, as it would make more

sense to talk about binding first and make the statement about the complex after.

It is an excellent suggestion. It has been implemented and the sentences swapped as suggested by the reviewer.

Lines 100-102. In Fig 2e, the color code for pink is defined "as scissile bond" yet all covalent bonds of the cytosine are colored like that. How about introducing a color for the actual scissile phosphate, or perhaps just have an arrow pointing to it?

We modified the figure and now the scissile bond is indicated with a pink arrow.

Lines 116: "paired end complex"? Which complex is this?

We replaced “paired end complex” with “IR-bound complexes” that is applicable to all the structures determined in complex with DNA.

Line 120: "In paired end complex, TnpA^{S911R}-IR100". Is the IR100 complex the paired end complex or is it one of the paired end complexes? Confusing.

To resolve the ambiguity, we modified this sentence to “In TnpA^{S911R}-IR100 complex...”

Lines 139-141: Here the authors attempt to address reviewer 3's comment, but again the result is not fully successful. "Theoretically reversible strand transfer reaction energetically favorable". First, the strand transfer reaction is not only theoretically reversible, it is actually reversible. As the reviewer pointed out, this reaction is neither favorable nor unfavorable as the substrate and the product are at about the same energy levels, so it can go back and forth on its own. This is different from the hydrolysis reaction of breaking the DNA chain that will not go backward on its own. How about: "...evoked for target bending which is to prevent the strand-transfer reaction reversing by moving the product 3'OH group sufficiently far away from the scissile phosphate".

We are thankful for the suggestion on improving the clarity of the manuscript. The proposed text modification is now implemented according to reviewer's #2 suggestion.

Line 143: "Didn't assemble into a canonical Shapiro intermediate". I have no idea what point the authors are trying to make. In the Shapiro intermediate, the broken target strand does not have the nts of the target site as correctly shown on Fig 1a.

We replaced Shapiro intermediate with “strand transfer product”.

Line 148: In Fig 2d, it is not really visible to which domain the disordered target density is adjacent to.

To improve the visibility, we colored the map surface corresponding to RNH domain in bordeaux in revised Fig. 2d. Now position of target density relative to the rest of the protein is clearer.

Line 160: How about "The lack of specificity of these interactions are permitting..." rather than "promiscuity"

We modified the text accordingly.

Lines 205, 208: I believe the authors meant to refer to Fig 5b not 4c.

Reference to the figures has been corrected.

Lines 211, 212: Can authors state that all hyperactive mutants impair immunity?

This is an interesting question. It is currently unclear whether TnpA hyperactivity strictly correlates with impairment in target immunity, but previous studies have shown that all characterized immunity mutants showed a higher propensity than wild-type TnpA to assemble the paired complex and to catalyze end cleavage in vitro. A statement has been added to the text to specify this.

To make the text unambiguous, we restricted the statement to characterized mutants: "Mapping other **characterized** hyperactive and target immunity deficient mutations ..."

Line 220: "The RNaseH fold insertion domain" I suggest a change: "The insertion domain that interrupts the RNaseH fold"

The text has been modified accordingly.

Line 262: It is not clear what distinguishes Tn5 among all other known transposase structures and what core is referred to. What numerical measures justify this choice?

To improve clarity here we modified the sentence as follows:

"Comparison of TnpA with other structurally characterized transposases revealed structural homology between TnpA's DBD4 and DBD of the cut-and-paste Tn5 transposase³⁰ (**Supplementary Table 3**) along with similarity in relative positions of RNH and DBD4 domains between these two transposases (**Supplementary Fig. 10c**)".

We refer to **Supplementary Table 3** which provides numerical comparison of structural similarity between these 2 transposases. Such a similarity was not observed for other transposases.

Lines 264-266: "This indicates..." I would suggest leaving this sentence out. "remarkably different evolution" is not something I can parse.

We modified the sentence to make the message clearer:

"This suggests a common evolutionary origin between these two otherwise structurally and mechanistically distinct transposases"

Lines 299-300: "...thread through the opening as a double-stranded break". I have no idea how a break would thread through and even if it could what would make such a break?

If we understand this correctly, the reviewer means that proposal that double stranded break threads through the opening in the protein, is pretty much impossible. And we completely agree with this. Nevertheless, following a pure logic, such option, however unconventional it may appear, cannot be completely excluded either, therefore we mentioned this scenario as a possible albeit unprovable option.

We have rephrased the sentence to make it clearer. Now it reads as follows: “Although we cannot formally exclude the possibility that DNA could be threaded through the protein from a preexisting double-strand break, it seems most likely that the TnpA dimer assembles onto the target prior or concomitantly to the formation of PEC.”

Lines 333-341: I think this section is not very clear. The authors are probably onto something, but it needs much clearer writing. What I can imagine is a scenario in which the target DNA, if it contained a Tn3 end, could be bound by the site-specific DBD domains. As they are now engaged with the target DNA, they will not be able to bind the real transposon end to be transposed. Perhaps the ability of the transposase to reconfigure itself to the extent as it seen in this work has also something to do with the reconfiguration needed to mediate immunity as well.

In response to reviewer’s #3 comments we modified this paragraph made the target immunity model clearer.

Lines 367-369: "highly concerted and regulated because...can damage...survival of the cell.." I think this is not the case. Tn3 transposition, given that it is targeted non-specifically, can damage anything it touches including the chromosome and can lead to cell death no matter how highly regulated it is. From the point of view of the transposon, there are certainly advantages of being regulated and concerted, but I would not mix in cell survival.

This is a valid argument. We have modified the sentence and removed ‘survival of the cell’. Now the sentence reads as follows:

“These reactions must be highly concerted and regulated because incomplete or abortive transposition can damage both the donor and the target molecule, thus compromising the survival of the transposon.”

Reviewer #3 (Remarks to the Author):

The authors have nicely strengthened and clarified the paper. My only significant remaining concern is the explanation for target immunity. After extensive pondering, here are my thoughts:

1) It is OK to just say that it is not fully understood. The system has many variables and to the best of my knowledge, nobody has managed to get full transposition to work with purified components, so we don’t even know what all the variables are.

We agree that our proposal is quite speculative, but its idea is drawn from the new structural data and the proposed mechanism is very simple. After carefully reading through the reviewer’s comments, we also adjusted the proposal for the mechanism to eliminate the weaknesses that our initial proposal had.

2) There is an apparent internal contradiction in the explanation between these two statements: “whenever transposon is already present in the proximal regions of the target, TnpA preferably binds transposon ends on the same DNA molecule” and “Therefore, the net probability of binding to transposon ends on another DNA molecule is higher producing an effect of reduced immunity.”

3) *The author's hypotheses would explain a preference for non-productive intramolecular (that is, intra-plasmid) transposition over intermolecular transposition, but that seems like an unlikely biological strategy for such a successful transposon. They don't explain why a target without transposon ends would be favored over one with transposon ends.*

4) *As a side point, the phrase "Transposition into the same DNA molecule results in the inversion or deletion of the transposon" needs re-wording to explain that it is the plasmid backbone that suffers inversions or deletions, not the transposon itself.*

5) *My best guess is that this system is conceptually similar to the Mu paradigm, but in this case those concepts are implemented very differently. Perhaps: (a) target-bound transpososomes are somehow dissociated when they encounter a single end; (b) this is in competition with (but faster than) the binding of two transposon ends (that end-binding is slow is supported by the not-fully-docked WT + transposon ends structure referred to, and the need for activating mutations); (c) this gives time to build up a differential distribution of target-bound transpososomes on targets that do not contain transposon ends. The overall concept is similar to the idea for Mu that before transpososomes are fully formed, end-bound MuA triggers dissociation of local MuB, such that by the time a proper Mu transpososome has formed, only distal targets are bound by MuB. In the Tn3-family case, exactly why target-bound transpososomes would tend to dissociate when they encounter a single transposon end would remain to be explained. Also, without an energy source my hypothesis would seem to be a Brownian ratchet that violates the laws of thermodynamics (https://en.wikipedia.org/wiki/Brownian_ratchet), but perhaps ATP-burning chaperones are involved.*

We are grateful to reviewer 3 for his/her efforts to implement our vision of target immunity. This was really inspiring. We believe that his/her proposition for target immunity mechanism is faithfully summarized in the following paragraph that we included in the manuscript to replace the previous description.

"The proposal that TnpA binds to the target DNA first suggests a plausible model for target immunity. The absence of DNA in the proposed target DNA binding site suggests that target DNA must have a specific conformation for binding or that the target DNA binding is the rate-limiting step of transpososome assembly. TnpA forms the active PEC on a fully assembled TnpA-target DNA complex in which the target DNA is enclosed and adequately positioned within the TnpA dimer (Fig. 6b). As described above, the formation of this complex requires that TnpA opens and closes around the target DNA, which may represent a slow step in the assembly process. Suppose the TnpA-target DNA assembles in the vicinity of a transposon. In that case, the interaction of TnpA with the transposon ends, prior to completion of target DNA binding, may lead to TnpA dissociation from or arrest of TnpA binding to the target DNA thus preventing the assembly of transpososome. Whenever TnpA associates with target on DNA regions remote from the transposon, the interference of TnpA-target DNA binding with transposon ends is reduced allowing for complete transpososome assembly."

6) *It seems easier to explain target immunity if the target binds 1st, and that agrees with the architecture of the transpososomes seen, but it does raise the question of why the authors*

don't see "target"-bound particles in which duplexes intended as transposon ends are instead being seen as target DNA.

This is a very good question, we believe that the target needs to have certain structural properties, that make it a permissive target. We see that some DNA molecules bind TnpA nonspecifically but there is no indication that they pass through the opening (**Supplementary Fig. 4c**). The target DNA once bound will be bent, possibly DNA bent during another process associated with DNA processing in the cell will be recognized as target DNA or DNA with a sequence prone to bending would be recognized as a target.

Miscellaneous small things:

1) *Figure 1 b abbreviates scaffold as SCF but the text uses SFD.*

We have fixed this by replacing the abbreviation in the main text with SCF.

2) *P146 – replace cysteine with cytosine*

Replaced.

3) *Lines 276 – 278: does it really falsify the previous work, or do the different results reflect different protein concentrations used in the different experiments? (that is, could mass action be driving dimerization in the structural work?)*

The dimerization domains interact over a surface of 1600 \AA^2 and the interactions are exclusively hydrophobic which is expected to create a tight binding between the protomers making a stable complex. The very rough estimate of the interaction energy is in the order of -30 kcal/mol (**Supplementary Table 4**) which corresponds to K_d of 10^{-22} M . Though very approximate, this number does suggest that the interaction mediated by the dimerization domains is very strong. The concentration of a single copy of TnpA in *E. coli* cell is around 1.5 nM (the concentration used for cryo-EM experiments is in order of 550 nM). These estimates suggest that under physiological conditions TnpA should exist as a stable dimer.

To make our statement less strong, we modified the sentence. Now it reads as follows:

“This observation contradicts the previously proposed model in which active transpososome assembles from TnpA monomers²⁹.”